# Brain IGF-I regulates LTP, spatial memory, and sexual dimorphic behavior

Raquel Herrero-Labrador[1,2,*], Joaquín Fernández-Irigoyen[3,4,*], Rebeca Vecino[1,2,*], Candela González-Arias[1], Karina Ausín[3], Inmaculada Crespo[1,2,5], Francisco J Fernández Acosta[1], Vanesa Nieto-Estévez[1,2], M José Román[1,2], Gertrudis Perea[1], Ignacio Torres-Alemán[1,2,6,†], Enrique Santamaría[3,4,†], Carlos Vicario[1,2]

Insulin-like growth factor-I (IGF-I) exerts multiple actions, yet the role of IGF-I from different sources is poorly understood. Here, we explored the functional and behavioral consequences of the conditional deletion of *Igf-I* in the nervous system (*Igf-I^{Δ/Δ}*), and demonstrated that long-term potentiation was impaired in hippocampal slices. Moreover, *Igf-I^{Δ/Δ}* mice showed spatial memory deficits in the Morris water maze, and the significant sex-dependent differences displayed by *Igf-I^{Ctrl/Ctrl}* mice disappeared in *Igf-I^{Δ/Δ}* mice in the open field and rota-rod tests. Brain *Igf-I* deletion disorganized the granule cell layer of the dentate gyrus (DG), and it modified the relative expressions of GAD and VGLUT1, which are preferentially localized to inhibitory and excitatory presynaptic terminals. Furthermore, *Igf-I* deletion altered protein modules involved in receptor trafficking, synaptic proteins, and proteins that functionally interact with estrogen and androgen metabolism. Our findings indicate that brain IGF-I is crucial for long-term potentiation, and that it is involved in the regulation of spatial memory and sexual dimorphic behaviors, possibly by maintaining the granule cell layer structure and the stability of synaptic-related protein modules.

## Introduction

Insulin-like growth factor-I (IGF-I) is a critical factor that regulates cell and tissue homeostasis at various levels, such as energy allocation, body and organ growth, life span, and tissue responses to stress (Liu et al, 1998; Pichel et al, 2003; Kappeler et al, 2008; Wu et al, 2009; Fernandez & Torres-Aleman, 2012; Bartke et al, 2013; Carlson et al, 2014; Nieto-Estevez et al, 2016a; Santi et al, 2018; Fernandez de Sevilla et al, 2022). In the central nervous system, the activities of IGF-I influence neural stem cell and progenitor cell proliferation (Arsenijevic et al, 2001; Aberg et al, 2003; Nieto-Estevez et al, 2016b); the survival, differentiation, and maturation of neurons, astrocytes and oligodendrocytes (Carson et al, 1993; Arsenijevic & Weiss, 1998; Cheng et al, 1998; Camarero et al, 2001; Vicario-Abejon et al, 2003, 2004; Otaegi et al, 2006; Cao et al, 2011; Corvin et al, 2012); the correct positioning of migrating neurons (Hurtado-Chong et al, 2009; Onuma et al, 2011; Li et al, 2012; Maucksch et al, 2013; Littlejohn et al, 2020); adult neurogenesis, synaptogenesis, neuron firing, and long-term potentiation (LTP) (Aberg et al, 2000; O'Kusky et al, 2000; Sun et al, 2005; Trejo et al, 2007; Llorens-Martin et al, 2010; Dyer et al, 2016; Hu et al, 2016; Nieto-Estevez et al, 2016a; Ogundele et al, 2018; Pristera et al, 2019; Maglio et al, 2021); and cognition, olfactory memory, spatial and skill learning, fear extinction memory, and anxiety (Trejo et al, 2007; Muller et al, 2012; Gazit et al, 2016; Liu et al, 2017; Frater et al, 2018; Farias Quipildor et al, 2019; Pristera et al, 2019; Zegarra-Valdivia et al, 2019; Maglio et al, 2021). The resulting net biological effect of IGF-I binding to the IGF-I receptor depends on the cell and tissue type, the age and sex of the animal, and the context or status in terms of stress (e.g., inflammation, neurodegeneration, neurovascular hypoxia or traumatic brain injury) (Gontier et al, 2015; Lopez et al, 2015; Nieto-Estevez et al, 2016a; Munive et al, 2016; Ashpole et al, 2017; De Magalhaes Filho et al, 2017; Gubbi et al, 2018; Santi et al, 2018; Cardoso et al, 2021).

Experiments on mice carrying a conditional deletion of *Igf-I* in the liver challenged the prevalent concept that blood IGF-I (mostly from the liver) was essential for normal postnatal growth. Rather, these studies appeared to emphasize the importance of locally synthetized IGF-I acting through autocrine and/or paracrine mechanisms (Sjogren et al, 1999; Yakar et al, 1999; Ohlsson et al, 2009; Wu et al, 2009). This latter concept was confirmed in more

[1]Instituto Cajal, Consejo Superior de Investigaciones Científicas (CSIC), Madrid, Spain   [2]CIBERNED, Instituto de Salud Carlos III (ISCIII), Madrid, Spain   [3]Proteored-ISCIII, Proteomics Platform, Navarrabiomed, Hospital Universitario de Navarra (HUN), Universidad Pública de Navarra (UPNA), Instituto de Investigación Sanitaria de Navarra (IdiSNA), Pamplona, Spain   [4]Clinical Neuroproteomics Unit, Navarrabiomed, Hospital Universitario de Navarra (HUN), Universidad Pública de Navarra (UPNA), Instituto de Investigación Sanitaria de Navarra (IdiSNA), Pamplona, Spain   [5]CES Cardenal Cisneros, Madrid, Spain   [6]Achucarro Basque Center for Neuroscience, and Ikerbasque Foundation for Science, Bilbao, Spain

Correspondence: cvicario@cajal.csic.es
*Raquel Herrero-Labrador, Joaquín Fernández-Irigoyen, and Rebeca Vecino are contributed equally to this study as first authors
†Ignacio Torres-Alemán and Enrique Santamaría are contributed equally to this study as principal investigators

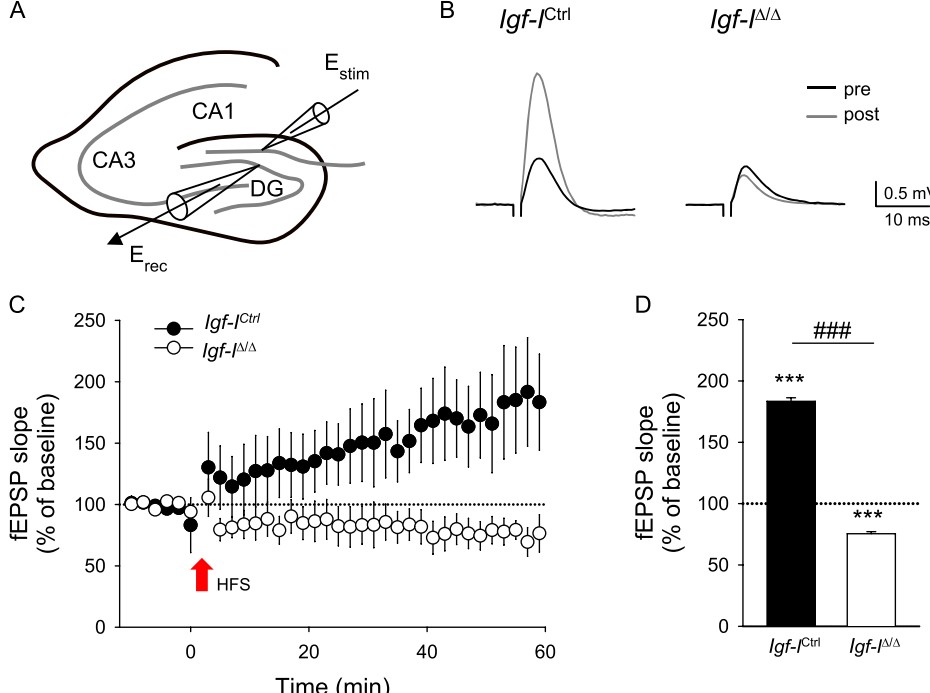

**Figure 1. Conditional deletion of brain insulin-like growth factor-I (IGF-I) abolishes hippocampal long-term potentiation.**
**(A)** Schematic representation of the experimental approach in hippocampal slices. The extracellular-recording pipette ($E_{rec}$) was place at the granular layer of dentate gyrus and to activate synaptic inputs, the stimulation electrode ($E_{stim}$) was located in the perforant pathway. **(B)** Representative average traces of field excitatory postsynaptic potential (fEPSP) recordings before (pre) and 60 min after (post) high-frequency stimulation (HFS) in *Igf-I*$^{Ctrl}$ and *Igf-I*$^{\Delta/\Delta}$ mice. **(C)** Relative fEPSP slope recordings (from basal values) from dentate gyrus hippocampal slices of 14–15 mo-old *Igf-I*$^{Ctrl}$ (black circles) and *Igf-I*$^{\Delta/\Delta}$ mice (white circles). The red arrow denotes the initiation of the HFS protocol. Note the absence of synaptic potentiation but also of significant depression of excitatory synaptic transmission in *Igf-I*$^{\Delta/\Delta}$ slices after HFS. **(D)** The average of the relative changes in the fEPSP slope 60 min after initiating HFS in slices from control (n = 3 mice, seven slices) and *Igf-I*$^{\Delta/\Delta}$ mice (n = 4 mice and seven slices), and the different responses to HFS of hippocampal slices from both mice. The results are represented as the mean ± SEM: ***$P < 0.001$ and $^{###}P < 0.001$, where "*" refers to the analysis in relation to the base line and "#" refers to the analysis between the two experimental groups.

recent studies using mice carrying a conditional *Igf-I* deletion in brain cells (Nieto-Estevez et al, 2016b; Pristera et al, 2019).

We previously found that brain IGF-I plays a major role in promoting the correct generation, migration, and maturation of neurons from neural stem cells during adult hippocampal neurogenesis (Nieto-Estevez et al, 2016b), although electrophysiological or behavioral phenotypes were not investigated in that study. Here, we show that the lack of brain IGF-I almost completely abrogates hippocampal LTP, and altering spatial memory, sex-dependent behavior, and granule cell layer (GCL) organization, and inducing major changes in the hippocampal proteome.

## Results

### Brain IGF-I is essential for hippocampal synaptic plasticity

The IGF-I signaling pathway fulfils a crucial role in the activity and plasticity of hippocampal synapses, and in hippocampus-dependent learning and memory (Trejo et al, 2007; Stern et al, 2014; Gazit et al, 2016), although the contribution of locally synthesized IGF-I to these processes remains to be fully elucidated. Here, we first evaluated the impact of local brain IGF-I deficiency on hippocampal synaptic plasticity, monitoring its potential consequences on LTP, a well-known cellular mechanism underlying learning and memory processes (Paulsen & Sejnowski, 2000). We recorded neuronal activity by measuring field excitatory postsynaptic potentials in the dentate gyrus (DG) before and after stimulating the perforant pathway using a high frequency stimulation (HFS) protocol (Fig 1A). HFS evoked robust synaptic potentiation in *Igf-I*$^{Ctrl}$ slices relative to the baseline (***$P < 0.001$, one way-ANOVA,

with post hoc Tukey's test: Fig 1B–D), yet synaptic potentiation was not evident in hippocampal slices from *Igf-I*$^{\Delta/\Delta}$ mice but rather, they displayed a significant depression of synaptic activity after the HFS protocol relative to the baseline (***$P < 0.001$, one way-ANOVA, with a post hoc Tukey's test: Fig 1B–D). The difference in the responses to HFS in the hippocampal slices from either mice was significant ($^{###}P < 0.001$: Fig 1D) and supports a critical role for brain-synthesized IGF-I in long-lasting synaptic plasticity at the hippocampal DG.

### Brain IGF-I regulates spatial memory and sexually dimorphic behavior

Because *Igf-I* depletion impairs hippocampal LTP, we tested whether two hippocampal-dependent tasks were affected in *Igf-I*$^{\Delta/\Delta}$ mice: spatial learning and memory (Figs 2 and 3). In the Morris water maze (MWM), *Igf-I*$^{\Delta/\Delta}$ mice displayed relatively higher escape latencies in the acquisition trials than the *Igf-I*$^{Ctrl/Ctrl}$ mice, yet these increases were only significant on day 4 (*$P < 0.05$, t test) when the mouse genotype alone was considered in the analysis (Fig 2). The escape latency also increased in *Igf-I*$^{\Delta/\Delta}$ females on day 3 (*$P < 0.05$, t test), whereas possible differences in *Igf-I*$^{\Delta/\Delta}$ males did not reach statistical significance ($P = 0.0509$ on day 2: Fig S1A and B).

We observed significant changes in the performance of *Igf-I*$^{\Delta/\Delta}$ mice relative to the *Igf-I*$^{Ctrl/Ctrl}$ mice during the transfer test when analyzed independently of the animal's sex (Fig 3A–C). These differences were evident through reductions in both the relative time spent in the platform (P) quadrant (**$P < 0.01$, t test: Fig 3B) and the relative distance swum in that quadrant (***$P < 0.001$, t test: Fig 3C). Moreover, the time spent by *Igf-I*$^{Ctrl/Ctrl}$ mice in each quadrant

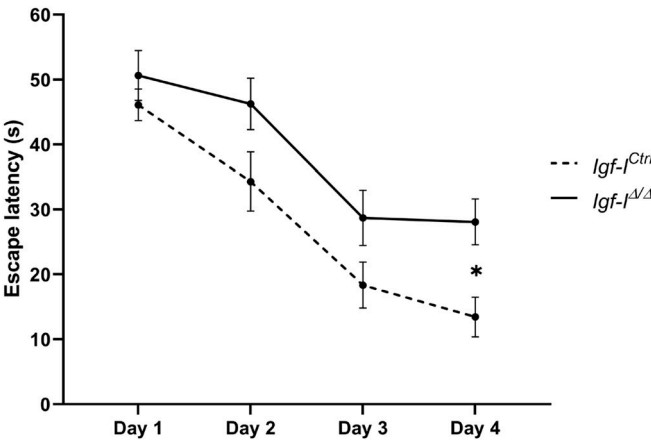

**Figure 2. The effect of brain insulin-like growth factor-I (IGF-I) deletion on spatial learning.**
Adult male and female *Igf-I*^Ctrl and *Igf-I*^Δ/Δ mice (6–16 mo old) were assayed in the water maze test. The escape latency showed a trend towards higher values in the *Igf-I*^Δ/Δ mice, which was significant at day 4. The results are the mean ± SEM (n = 9–11 mice). *P < 0.05.

differed significantly (^#P < 0.05, ^###P < 0.001, ^####P < 0.0001, one-way ANOVA followed by Tuckey post hoc test: Fig 3B), whereas *Igf-I*^Δ/Δ mice spent a similar time in all four quadrants (Fig 3B). In accordance with this, the distance swum by *Igf-I*^Ctrl/Ctrl mice in each quadrant differed significantly (^##P < 0.01, ^###P < 0.001, ^####P < 0.0001, one-way ANOVA followed by a Tuckey post hoc test: Fig 3C).

The absolute time spent by *Igf-I*^Ctrl/Ctrl mice in each quadrant also differed significantly (^#P < 0.05, ^##P < 0.001, ^####P < 0.0001, Welch's ANOVA test followed by Games–Howell post hoc test Fig S2A), whereas *Igf-I*^Δ/Δ mice spent a similar time in all four quadrants. By contrast, the swimming speed in the pool (Fig S2C) and the mean velocity in each quadrant were largely unaffected by *Igf-I* deletion (Fig S2C'). Finally, no changes in the total freezing time were observed in the four quadrants (Fig S2D). However, there were significant differences in the *Igf-I*^Ctrl/Ctrl mice when the relative freezing time was compared between quadrants (^#P < 0.05, ^##P < 0.01, Welch's ANOVA test followed by Games–Howell post hoc test: Fig S2D'). Together, these findings suggest that brain IGF-I plays a crucial role in the formation of new spatial memories.

A series of analyses were then performed to determine whether the animal's sex might influence the spatial memory of *Igf-I*^Ctrl/Ctrl and *Igf-I*^Δ/Δ mice. Indeed, female mice appeared to spend relatively longer in the P quadrant than males, although only the difference between *Igf-I*^Δ/Δ female and male mice was significant (*P < 0.05, t test: Fig 3D). In addition, the *Igf-I*^Δ/Δ females spent relatively less time in the P quadrant than the *Igf-I*^Ctrl/Ctrl females (^#P < 0.05, t test: Fig 3D). *Igf-I*^Ctrl/Ctrl female mice swam a greater relative distance in quadrant P than the *Igf-I*^Ctrl/Ctrl males (*P < 0.05, t test: Fig 3E). Moreover, the relative distance swum in quadrant P by both *Igf-I*^Δ/Δ male and female mice was significantly less than that swum by *Igf-I*^Ctrl/Ctrl male and female mice (^##P < 0.01, t test: Fig 3E).

When the relative time spent by the mice in the different quadrants was plotted according to sex, more significant changes were observed among the *Igf-I*^Ctrl/Ctrl mice in females than in males (^#P < 0.05; ^###P < 0.001; ^####P < 0.0001, one-way ANOVA followed by Tuckey

post hoc test: Fig S2A'). In addition, *Igf-I*^Δ/Δ females spent longer in the P quadrant than *Igf-I*^Δ/Δ male mice (*P < 0.05, t test: Fig S2A"). Some similarities were observed in the relative distance swum in the different quadrants, with the most significant changes observed in the female as opposed to the male *Igf-I*^Ctrl/Ctrl mice (^#P < 0.05; ^####P < 0.0001, one-way ANOVA followed by Tuckey post hoc test: Fig S2B), although these significant differences were not evident in the *Igf-I*^Δ/Δ mice (Fig S2B'). Moreover, *Igf-I*^Ctrl/Ctrl female swam longer distances in the P quadrant than *Igf-I*^Ctrl/Ctrl male mice (*P < 0.05, t test: Fig S2B). By contrast, no significant changes in the total velocity (Fig S2C") or in the total freezing time (Fig S2D") were observed between the male and female mice. These results indicate that the effect of brain IGF-I on spatial memory formation is at least in part sex-dependent.

Additional phenotyping of the IGF-I-deficient mice (*Igf-I*^Δ/Δ) showed a loss of sexual dimorphism in various behaviors (Fig 4). Horizontal and vertical deambulatory activity was determined in the open field test (OFT) and it was similar in both sexes of *Igf-I*^Δ/Δ mice, whereas it differed significantly in male and female *Igf-I*^Ctrl littermates (*P < 0.05, two way-ANOVA, with post hoc Bonferrani's test: Fig 4A and B). The OFT can also be used to assess stress by measuring stereotypic movements and the time spent in the center of the arena. *Igf-I*^Δ/Δ mice did not show sexual dimorphism for either of these parameters, whereas this dimorphism was evident in control mice (*P < 0.05 and **P < 0.01, two way-ANOVA, with post hoc Bonferrani's test: Fig 4C and D). A similar loss of sexual dimorphism between male and female *Igf-I*^Δ/Δ mice was evident in terms of their motor coordination measured in the rota-rod test (Fig 4E). By contrast, no significant changes were observed in the elevated plus maze that assesses anxiety-related behavior (Fig S3).

## Hippocampal proteome and the structure of the GCL rely on brain IGF-I

To examine the consequences of IGF-I deficiency on hippocampal homeostasis, we applied shotgun proteomics to obtain novel information about the site-specific molecular signature of male and female *Igf-I*^Δ/Δ mice in the hope that this may help explain the electrophysiological and behavioral phenotypes observed. As such, hippocampal proteostasis was monitored using tandem mass tags (TMTs) coupled to tandem mass spectrometry (MS), detecting 65 differentially expressed proteins (DEPs) in female *Igf-I*^Δ/Δ mice and 58 DEPs in male *Igf-I*^Δ/Δ, relative to their respective *Igf-I*^Ctrl mice (Fig 5A and Table S1). No proteins were deregulated uniformly across both cohorts. However, most of the DEPs mapped to vesicle-related compartments, such as the TGN, presynapses, vesicle tethering complexes, and the early endosome (Fig 5B), with membrane trafficking the only biological activity disrupted at the protein level in both the female and male *Igf-I*^Δ/Δ hippocampi. This conclusion was also reached by performing a bioinformatics analysis that highlighted different functional profiles between the sexes of these mice (Fig S4).

Interlocking the differentially expressed proteomes with the SYNGO and Ingenuity Pathway Analysis repositories revealed alterations to synaptic proteins and to proteins involved in LTP. Specifically, *Igf-I* deficiency in males modulated a synaptic protein module comprised of FLOT1, RAB8A, RAB2A, BAIAP2, PLCB3, RPL7A, LGI1, PENK, CTNNA2, ADGRL1, VPS45, RHEB, FBXO2, PICK1, NF1, and ATP6AP1, whereas *Igf-I*^Δ/Δ female mice displayed changes in expression of a quite different set of synaptic

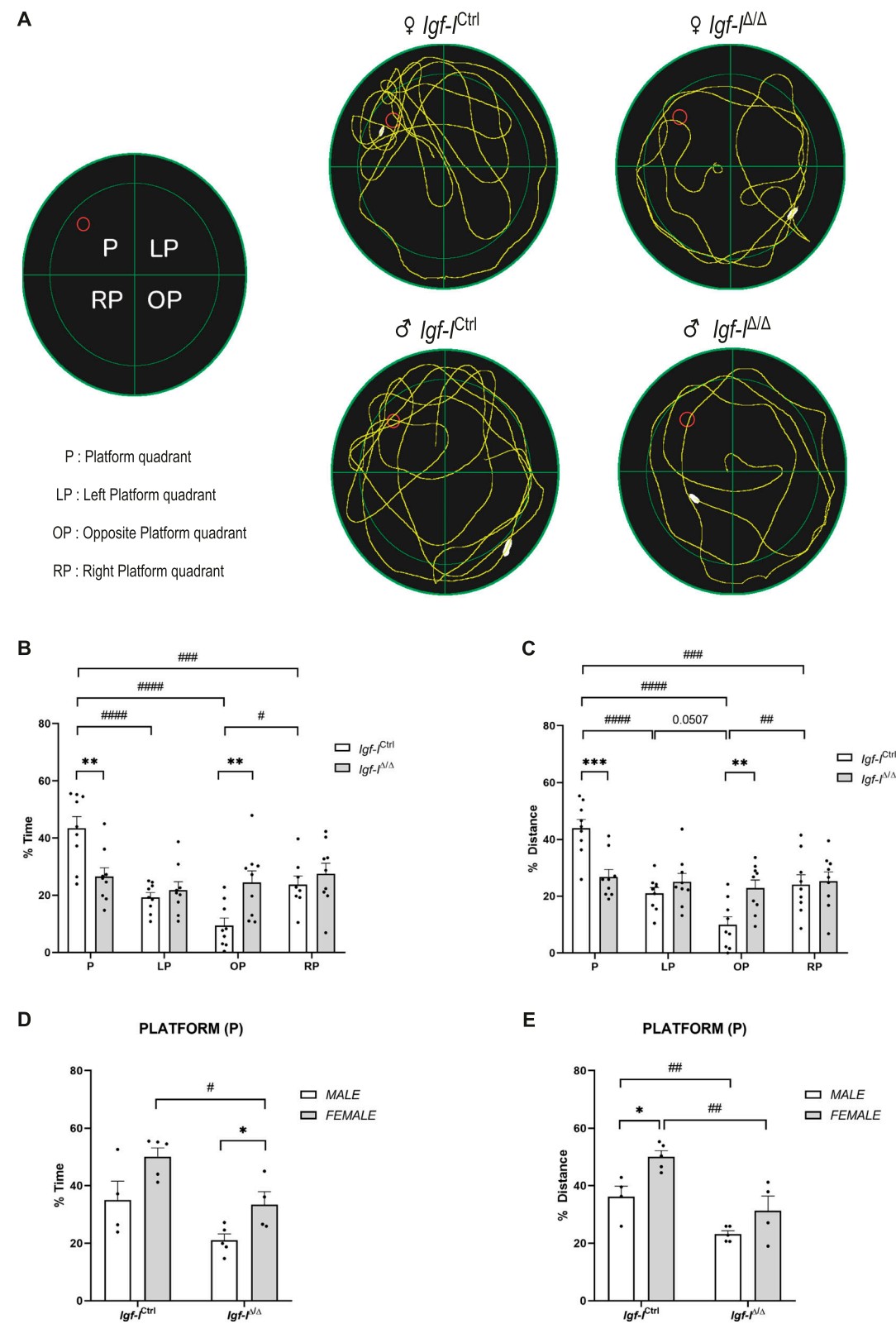

**Figure 3. Conditional deletion of brain insulin-like growth factor-I (IGF-I) produces spatial memory deficits.**
Adult male and female *Igf-I*^Ctrl and *Igf-I*^Δ/Δ mice (6–16 mo old) were assayed in the Morris water maze test. **(A)** The images show representative traces of the routes swum by the different mice in the four quadrants of the pool (P, LP, OP, and RP). **(B, C)** Significant differences between *Igf-I*^Δ/Δ and *Igf-I*^Ctrl mice were observed in the relative time spent (**P < 0.01 (B)) and relative distance swum (***P < 0.001: (C)) in the platform quadrant (P). **(B)** As seen from the graphs, *Igf-I*^Δ/Δ mice spent a similar relative time in the

proteins that included PRKCE, RAB6A, CASK, VAMP7, ERC1, GPHN, RPL6, MYO6, CTNND2, KCND2, SH3GL3, LYN7C, and NBEA (Table S2).

These alterations were accompanied by changes in protein intermediates of LTP and long-term depression, such as PAFAH1B2 and PRKCE in $Igf-I^{\Delta/\Delta}$ females, and BAIAP2/IRSp53 together with PPP2R5A in $Igf-I^{\Delta/\Delta}$ males. We analyzed the relationships between the altered hippocampal proteomes and the estrogen/androgen pathways or neurogenesis using the STRING and BIOGRID tools (Szklarczyk et al, 2019; Oughtred et al, 2021). Proteins involved in estrogen and androgen metabolisms were functionally connected to some of the deregulated hippocampal proteins (Fig 6). Furthermore, according to the BioGrid repository, 25% of the altered hippocampal proteins in female $Igf-I^{\Delta/\Delta}$ mice have been previously characterized as proteins that physically interact with the oestrogen receptors ESR1 (16 proteins—MYO6, AGFG1, KHSRP, CEP170B, CKAP5, COX4I1, LIN7C, LUZP1, MINK1, PDK3, TPP1, RPL6, CASK, CTNND2, PSMC1, and NDUFB3) or ESR2 (17 proteins—MYO6, AGFG1, KHSRP, CEP170B, CKAP5, COX4I1, LIN7C, LUZP1, MINK1, PDK3, TPP1, CTSB, EZR, MCCC2, UGGT1, SFXN1, and PSMA5). Significantly, none of the hippocampal proteins altered in male $Igf-I^{\Delta/\Delta}$ mice is currently considered to interact with androgen receptors.

To complement our study, we wanted to determine whether we could obtain functional evidence of a relationship between genes involved in neurogenesis (Nieto-Estevez et al, 2016b) and the dyshomeostatic proteome detected at the level of the hippocampus in $Igf-I^{\Delta/\Delta}$ mice. As such, both the differential proteomes were interlocked with HES5, NEUROG2, CALB1, and DSCAML1, and although no functional relationships were observed in male $Igf-I^{\Delta/\Delta}$ mice, deregulated proteins were identified in female $Igf-I^{\Delta/\Delta}$ mice that were linked to CALB1, such as CASK1, GPHN, and KCND2 (Fig 7).

To search for specific targets affected by $Igf-I$ deletion, we decided to investigate changes in individual hippocampal cells and proteins in $Igf-I^{\Delta/\Delta}$ mice. Because we previously found that $Igf-I$ deletion affected the positioning of Prox1+ neurons in the GCL of postnatal day 49 $Igf-I^{\Delta/\Delta}$ mice (Nieto-Estevez et al, 2016b), we sought to investigate whether this phenotype was maintained in old mice (6–16 mo old). Immunohistochemistry (IHC) with an antibody against Prox1 identified more ectopic Prox1+ neurons (arrowheads) in the molecular layer (ML, 2.5-fold, *$P$ < 0.01, $t$ test: Fig 8A and B) and hilus (Hi, 2.5-fold, *$P$ < 0.05, $t$ test: Fig 8A and C) of $Igf-I^{\Delta/\Delta}$ than in $Igf-I^{Ctrl/Ctrl}$ mice. Therefore, local synthesis of IGF-I is necessary for the correct positioning of granule cells in the GCL throughout the animal's life.

Having seen that $Igf-I$ deletion altered the hippocampal proteome, we assessed whether this deletion produces changes in the relative expression of specific proteins in Western blots and using IHC. There was less RAB6A in the hippocampus of $Igf-I^{\Delta/\Delta}$ mice, a small GTPase involved in LTP regulation (Gerges et al, 2005; Hausser & Schlett, 2019), although the change relative to $Igf-I^{Ctrl/Ctrl}$ mice did not appear to be significant ($P$ = 0.0554, $t$ test: Figs 9A and S5A). A similar trend was observed for RAB2A (Gerges et al, 2005; Hausser &

Schlett, 2019), gephyrin (Choii & Ko, 2015; Ravasenga et al, 2022) and VAMP7 (Kandachar et al, 2018), although the data dispersion was high and the small reductions observed in $Igf-I^{\Delta/\Delta}$ mice were not significant (Figs 9B–D and S5A–C). Moreover, no differences were detected between males and females (Figs 9A–D and S5A–C).

To investigate whether RAB6A might be specifically modulated in neurons rather than in the hippocampus as a whole, we immunostained sections with antibodies against RAB6A and MAP2ab, the latter a general marker of neurons (Vicario-Abejon et al, 1998). We evaluated the co-localization of both these proteins in neurons located in the Hi, as they were easily identified (Fig S6). The fluorescence intensity was expressed in relative fluorescence units (RFU), and it was similar in both $Igf-I^{\Delta/\Delta}$ and $Igf-I^{Ctrl/Ctrl}$ mice. Moreover, no differences were found in the proportion of neurons immunoreactive for both MAP2 and RAB6A relative to the total number of MAP2+ neurons in the Hi. Similarly, no change in the RFU was evident when sections from $Igf-I^{\Delta/\Delta}$ and $Igf-I^{Ctrl/Ctrl}$ mice were immunostained with an antibody against the postsynaptic protein gephyrin (Fig S7), and against Synapsin-I and PSD95, a pre and a postsynaptic protein, respectively (Fig S8) (Vicario-Abejon et al, 2002), or an antibody against aromatase (an enzyme that catalyses oestrogen synthesis from androgens, Fig S9) (Garcia-Segura et al, 2010). By contrast, we detected significant changes when hippocampal sections from $Igf-I^{\Delta/\Delta}$ and $Igf-I^{Ctrl/Ctrl}$ mice were immunostained with antibodies against GAD65 and VGLUT1, two markers of inhibitory and excitatory neurons, respectively (Vicario-Abejon et al, 2002). In fact, GAD65 RFU was 36% higher in $Igf-I^{\Delta/\Delta}$ than in $Igf-I^{Ctrl/Ctrl}$ mice (*$P$ < 0.05, $t$ test: Fig 10A and B), whereas VGLUT1 RFU was a 12% higher in $Igf-I^{\Delta/\Delta}$ than in $Igf-I^{Ctrl/Ctrl}$ mice (*$P$ < 0.05, $t$ test: Fig 10A and C), suggesting that $Igf-I$ deletion could provoke an imbalance in the inhibitory/excitatory ratio in the DG. Because parvalbumin (PVA) expressing neurons in the Hi drive synaptic inhibition in DG granule cells (Afrasiabi et al, 2022), we assessed the distribution of PVA in sections using an antibody raised against this protein (Fig 11). However, although the average number of PVA+ neurons in the Hi, ML, and GCL were all higher in $Igf-I^{\Delta/\Delta}$ than $Igf-I^{Ctrl/Ctrl}$ mice, these increases were not significant.

Altogether, these findings suggest that $Igf-I$ deletion alters the structure of the GCL, the inhibitory/excitatory ratio, and hippocampal protein modules, which could ultimately lead to impairing LTP, producing deficits in spatial memory formation and affecting sexual dimorphic behaviors.

## Discussion

Elucidating the role of circulating (largely derived from the liver) and locally synthesized IGF-I both during brain maturation and in its functioning has been the focus of many studies, although this issue remains unresolved. Early studies suggested

four quadrants, whereas the relative time spent by $Igf-I^{Ctrl}$ mice varied between quadrants (#$P$ < 0.05, ###$P$ < 0.001, ####$P$ < 0.0001: (B)). In line with this, $Igf-I^{\Delta/\Delta}$ mice swam a similar relative distance in the four quadrants, whereas the relative distance swum by $Igf-I^{Ctrl}$ mice varied between the quadrants (##$P$ < 0.01, ###$P$ < 0.001, ####$P$ < 0.0001). **(D, E)** Female $Igf-I^{\Delta/\Delta}$ mice spent significantly less relative time in the P quadrant (*$P$ < 0.05: (D)), whereas both female and male $Igf-I^{\Delta/\Delta}$ mice swam less distance in the P quadrant (##$P$ < 0.01: (E)). **(D, E)** Both the relative time spent and the relative distance swum displayed a sexual dimorphic pattern when comparing $Igf-I^{\Delta/\Delta}$ and $Igf-I^{\Delta/\Delta}$ male and female mice (*$P$ < 0.05: (D)), and when comparing $Igf-I^{Ctrl}$ and $Igf-I^{Ctrl}$ male and female mice (*$P$ < 0.05: (E)). The results are the mean ± SEM (n = 4–9 mice).

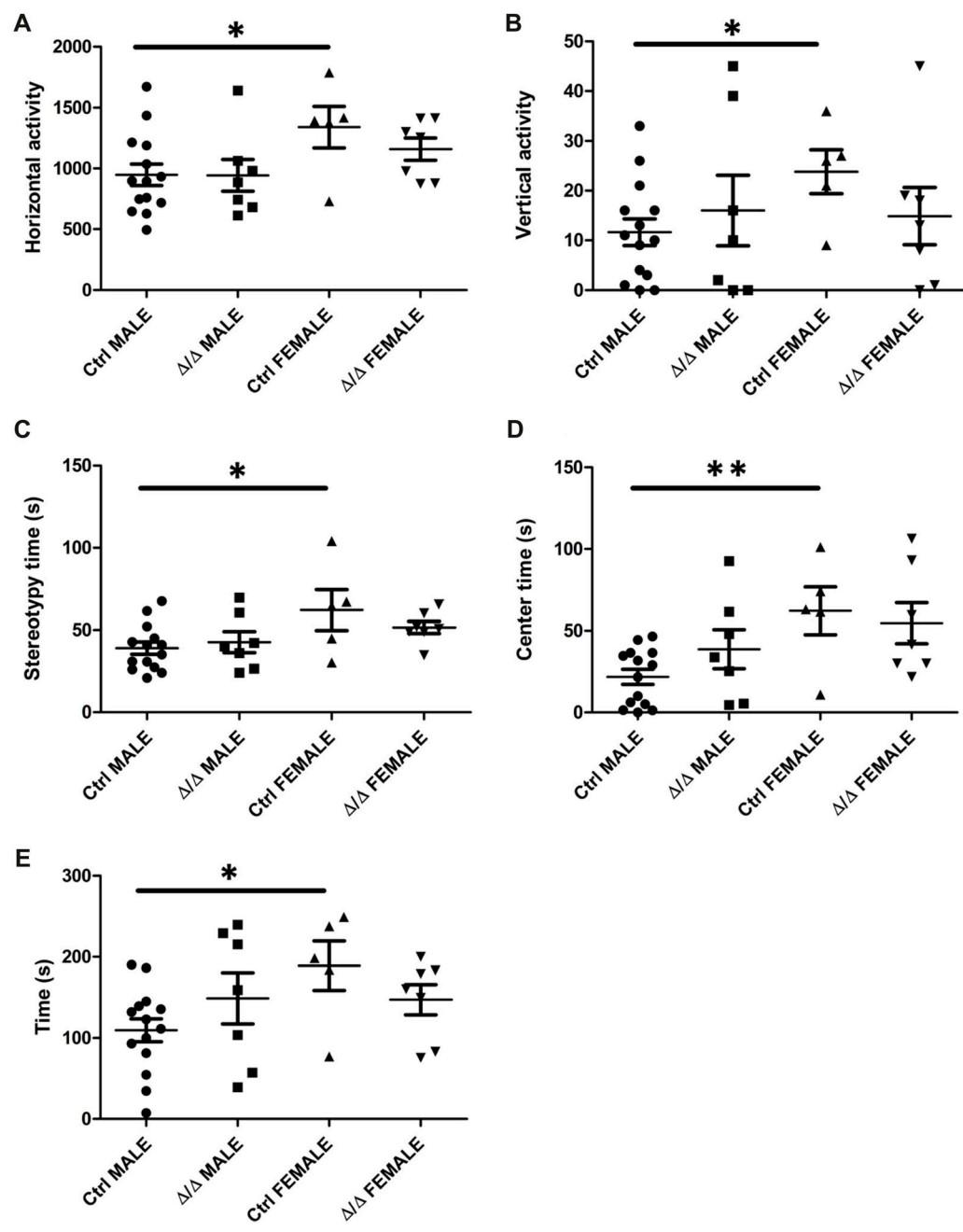

**Figure 4. Conditional deletion of brain insulin-like growth factor-I (IGF-I) alters sexual dimorphic behaviors.**
**(A, B, C, D)** Conditional deletion of brain IGF-I produces a loss of sexual dimorphism in the open field test. **(A, B, C, D)** Adult male and female *Igf-I*^Ctrl and *Igf-I*^Δ/Δ mice (6–10 mo old) were assayed in the open field test, and the significant differences between male and female *Igf-I*^Ctrl mice were found in deambulatory horizontal activity (A), vertical activity (B), stereotypic activity (C), and the time spent in the center of the arena (D) were lost in *Igf-I*^Δ/Δ mice. **(E)** Conditional deletion of brain IGF-I produces a loss of sexual dimorphism in motor coordination. The significant differences found in the time spent on the rota-rod between male and female *Igf-I*^Ctrl mice were not evident in *Igf-I*^Δ/Δ mice. The data are represented as the means ± SEM (n = 5–14 mice per condition): *$P < 0.05$.

that most of IGF-I's activities were carried out by circulating IGF-I, acting as an endocrine hormone on brain cells (Russo et al, 2005; Ohlsson et al, 2009; Wu et al, 2009; Fernandez & Torres-Aleman, 2012). However, the finding that mice deficient in liver *Igf-I* develop a relatively mild phenotype (Sjogren et al, 1999; Yakar et al, 1999) prompted studies into the autocrine and/or paracrine role of locally synthesized IGF-I. Nonetheless, it has not been easy to

address this possibility because of the lack of suitable animal models.

Using a conditional *Igf-I* knockout in which brain IGF-I synthesis is severely diminished (Nieto-Estevez et al, 2016b), we show that this growth factor is essential for hippocampal LTP and the acquisition of new spatial memories, and that it regulates sexual dimorphic behaviors. Furthermore, brain IGF-I is critical to maintain

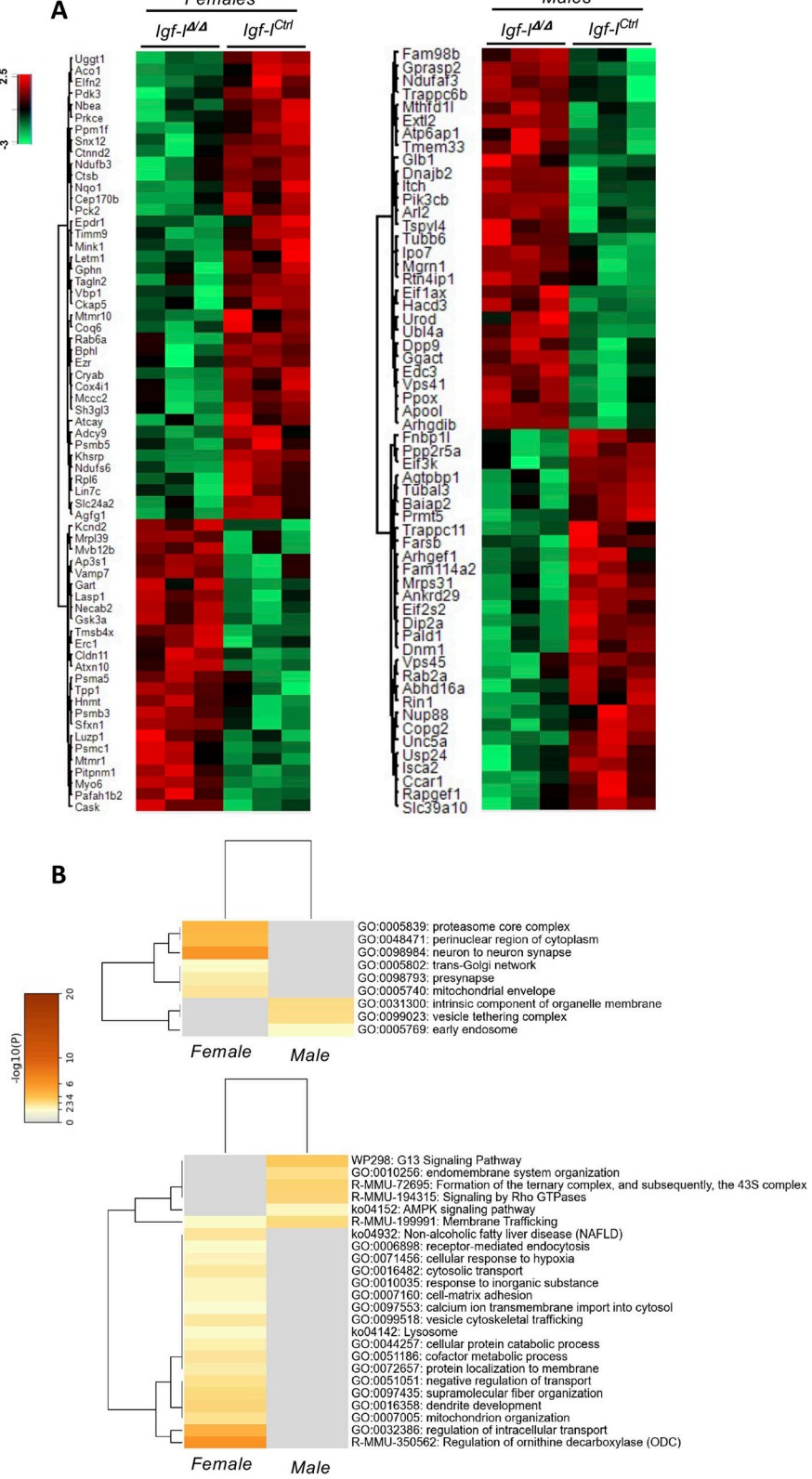

**Figure 5. Hippocampal protein homeostasis is disrupted by insulin-like growth factor-I (IGF-I) deficiency.**

**(A)** Heatmap representation showing both the clustering and the degree of change for the differentially expressed proteins (DEPs) in the hippocampus of 10–15 mo-old female (65 DEPs) and male (58 DEPs) $Igf-I^{\Delta/\Delta}$ mice relative to the $Igf-I^{Ctrl}$ mice (n = 3 mice per condition). Note that the same proteins were not deregulated in mice of both sexes. **(B)** Functional clustering of DEPs in the hippocampus of female and male $Igf-I^{\Delta/\Delta}$ mice relative to the $Igf-I^{Ctrl}$ mice according to the subcellular distribution (upper) and pathway mapping (lower).

**A**

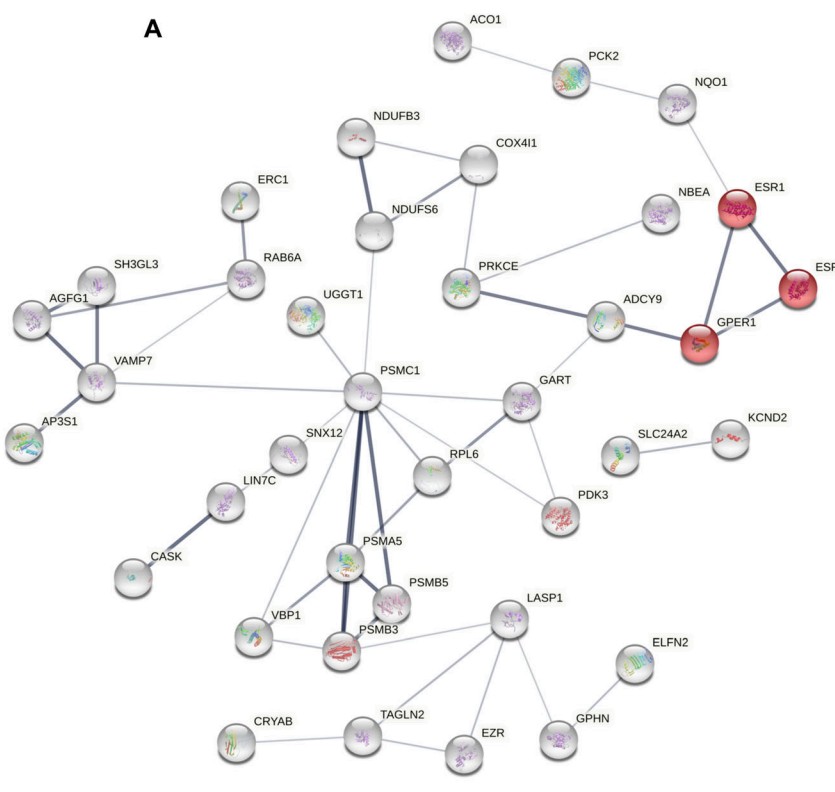

**Figure 6. Functional interactomes of the deregulated hippocampal proteomes in *Igf-I*^Δ/Δ mice.**
**(A)** Functional relationships between differentially expressed proteins in female *Igf-I*^Δ/Δ mice with protein intermediates involved in estrogen metabolism (ESR1, ESR2, and GPER1: highlighted in red), including ADCY9 and NQO1. **(B)** Functional relationships between differentially expressed proteins in male *Igf-I*^Δ/Δ mice with protein intermediates involved in androgen metabolism (SRD5A1, SRD5A2, and SRD5A3: highlighted in red), including CCAR1, PRMT5, and UBL4A. Protein interactomes were constructed using the STRING tool: ESR1 and ESR2, estrogen receptors 1 and 2; GPER1, G protein-coupled estrogen receptor 1; SRD5A1, SRD5A2, and SRD5A3, steroid-5 alpha reductases 1, 2, and 3.

**B**

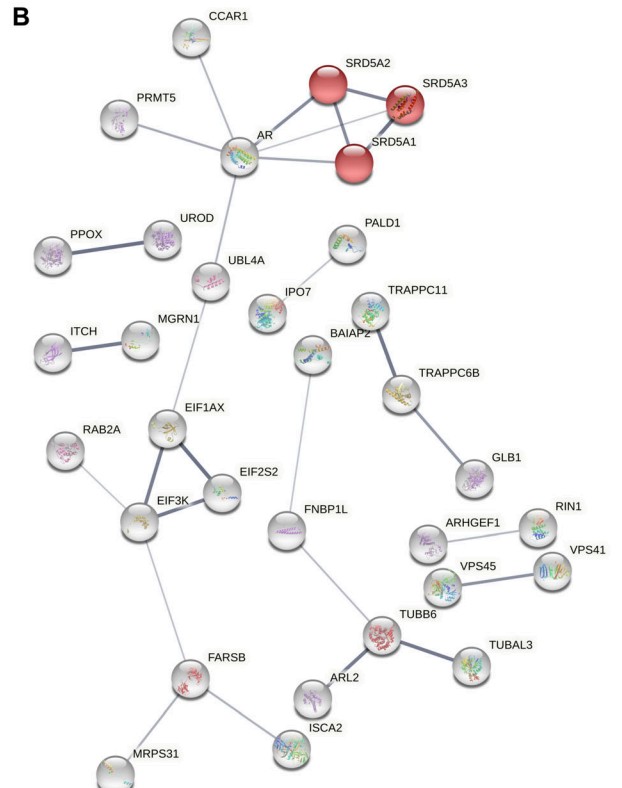

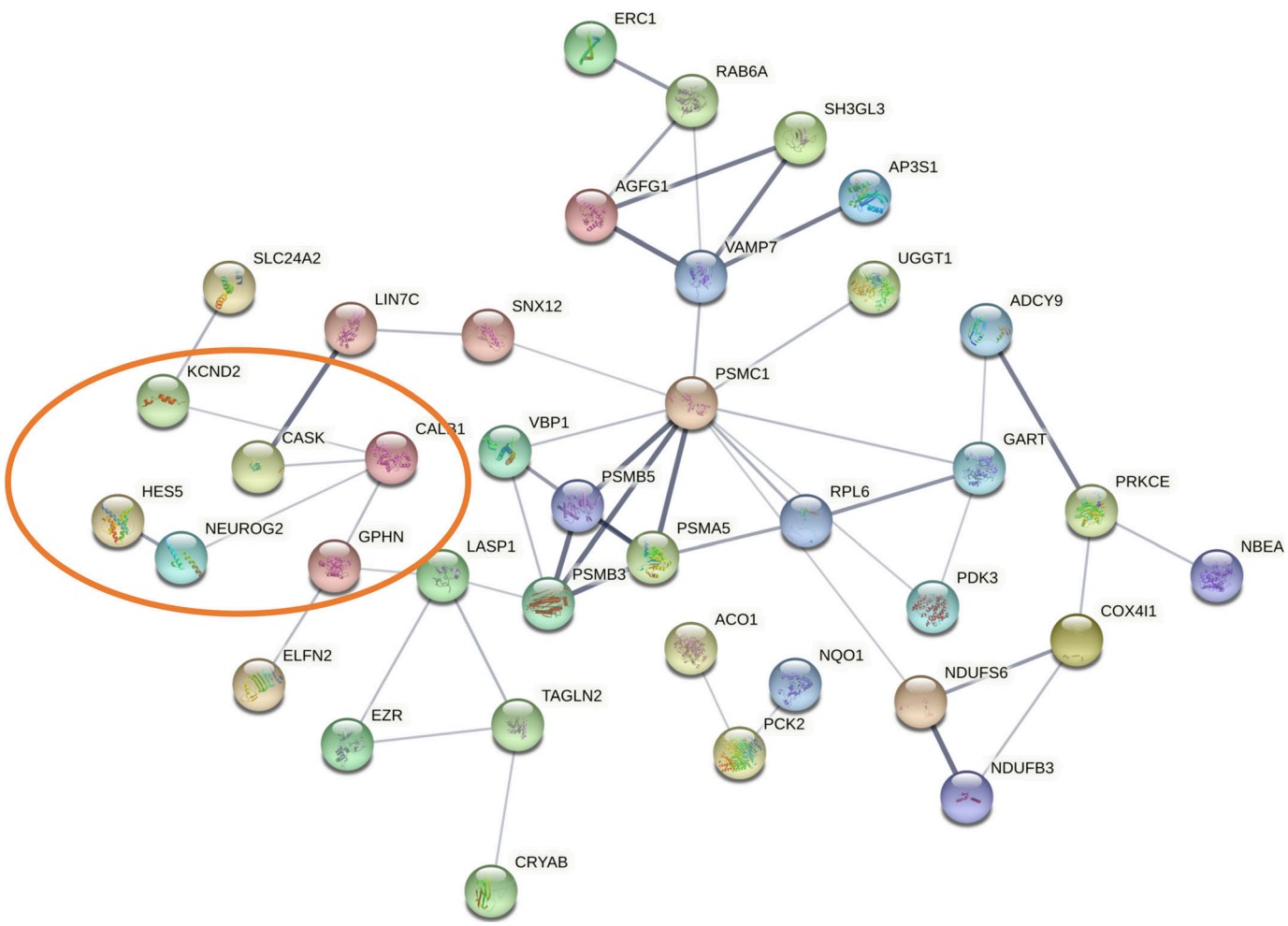

**Figure 7. Functional relationships between proteins involved in neurogenesis and the dysregulated hippocampal proteome of female *Igf-I*<sup>Δ/Δ</sup> mice.**
The orange circle indicates specific connections with calbindin (CALB1), and they include CASK1, NEUROG2, and gephyrin (GPHN). No functional relationships were observed in the male *Igf-I*<sup>Δ/Δ</sup> mice.

the hippocampal proteome, the cellular organization of the DG (particularly that of the GCL), and possibly, the balance between inhibitory and excitatory synaptic activities.

A crucial finding here is that the loss of brain *Igf-I* impairs the formation of new spatial memories, which links the failure to express LTP with a hippocampal-related behavior (Paulsen & Sejnowski, 2000; Lynch, 2004). It is important to emphasize that there is a strong reduction in the *Igf-I* expression in the brains of the *Igf-I*<sup>Δ/Δ</sup> mice studied here although these animals have normal serum IGF-I levels, supporting the conclusion that the loss of LTP and the impaired spatial memory are because of the lack of locally synthesized IGF-I. Because a liver IGF-I deficiency disrupted both hippocampal LTP and spatial memory (Trejo et al, 2007), our study adds further insights to the regulation of these processes by IGF-I via autocrine and/or paracrine mechanisms. This could be critical to link activity-dependent IGF-I synthesis in the hippocampus with synaptic plasticity and with the formation of new spatial memories.

Notably, female and male *Igf-I*<sup>Δ/Δ</sup> mice have a completely distinct set of DEPs, with membrane trafficking being the only biological activity commonly disrupted in the hippocampus of

both sexes. Moreover, alterations to synaptic proteins occur in both female and male mice, and proteins involved in LTP are also altered. These include the small GTPase Rab6A in females and Rab2A in males, both of which play important roles in regulating ionotropic AMPA glutamate receptor trafficking from the Golgi apparatus to the postsynaptic membrane during LTP (Gerges et al, 2005; Hausser & Schlett, 2019). In this regard, less RAB6A was detected in Western blots of the *Igf-I*<sup>Δ/Δ</sup> hippocampus, although apparently not significantly P = 0.0554. In addition, our bioinformatics analysis reveals a number of pre and postsynaptic proteins whose function could be altered, affecting synaptic activity in both males and females. However, no changes were observed in gephyrin, VAMP7, synapsin-I or PSD95. By contrast, we observed changes in the relative expressions of GAD and VGLUT1, which are preferentially localized to inhibitory and excitatory presynaptic terminals, respectively (Vicario-Abejon et al, 2002). Because *Igf-I*<sup>Δ/Δ</sup> mice appear to have relatively stronger GAD staining than that of VGLUT1, it is tempting to hypothesize that there is an imbalance in the inhibitory to excitatory ratio in the *Igf-I*<sup>Δ/Δ</sup> hippocampus that could affect LTP and synaptic plasticity. In

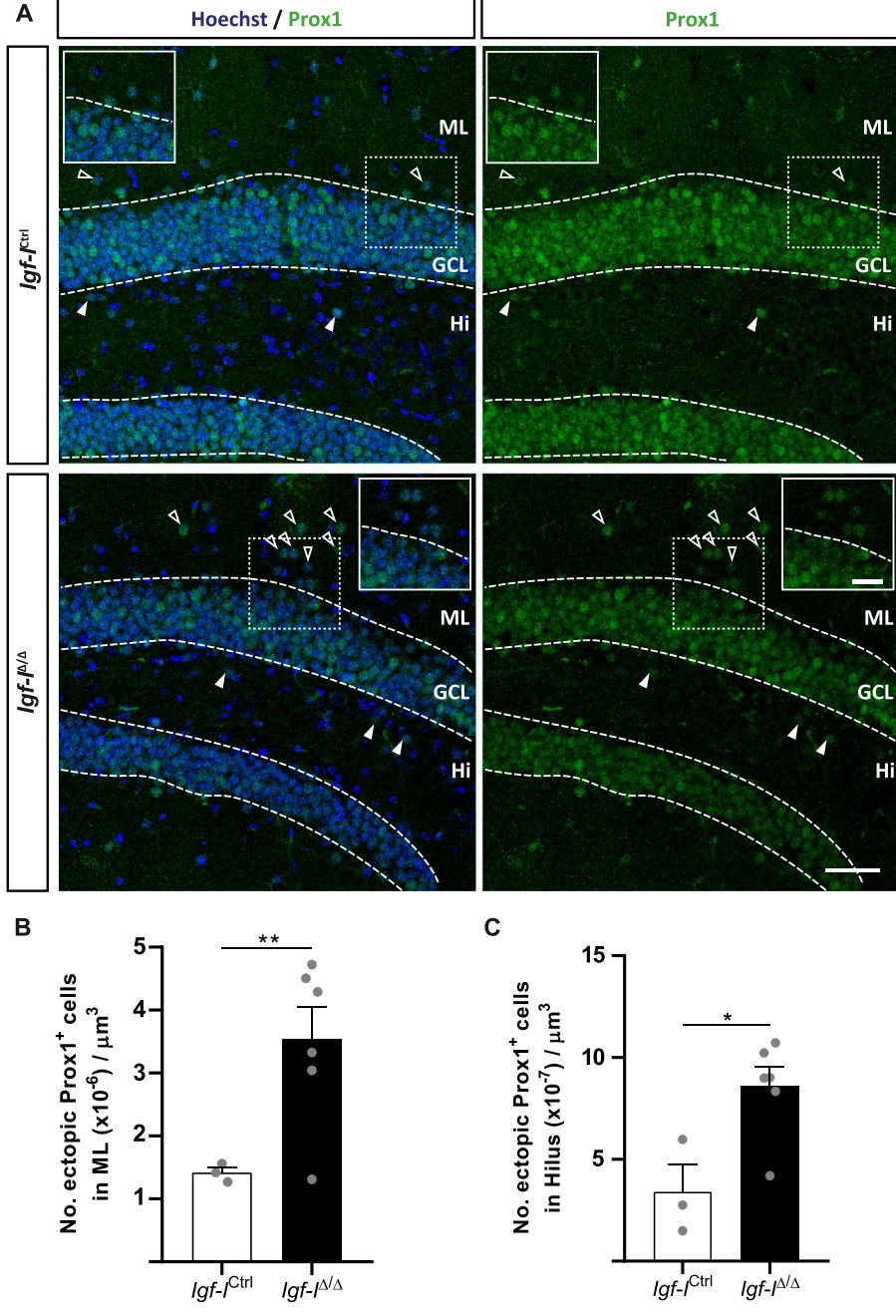

**Figure 8. Brain insulin-like growth factor-I (IGF-I) deletion alters the structure of the granule cell layer in the dentate gyrus.**
**(A)** Representative images taken from hippocampal sections immunostained with an anti-Prox1 antibody and with the nuclei stained with Hoechst.
**(B, C)** The arrowheads indicate ectopic Prox1+ cells, and the number of ectopic Prox1 cells was significantly higher in the molecular layer and in the hilus (Hi) of $Igf-I^{\Delta/\Delta}$ mice (see the graphs in (B, C), respectively). The results are the mean ± SEM (n = 3 and six mice): *$P < 0.05$, **$P < 0.01$. Scale bar = 50 $\mu$m; insets = 20 $\mu$m.

addition, our data suggest that the disruption of synaptic protein modules may also underlie the failure to induce hippocampal LTP in the $Igf-I^{\Delta/\Delta}$ hippocampus. Furthermore, the GCL of $Igf-I^{\Delta/\Delta}$ mice is disorganized because of the presence of ectopic Prox1+ neurons in the ML and Hi. Because the recording electrode in such studies is placed in the GCL, this disorganization, together with the impaired dendritic complexity and differentiation of newly born granule neurons and the reduced size of the DG (Nieto-Estevez et al, 2016b), could potentially affect the efficacy of the synaptic network in response to HFS. By contrast, elevating IGF-I levels in the brain enhances neurogenesis and neuron survival, and it

promotes synaptic plasticity by regulating AMPA glutamate receptor phosphorylation and trafficking (Nishijima et al, 2010; Fernandez & Torres-Aleman, 2012; Littlejohn et al, 2020; Williams et al, 2022).

Notably, these processes appear to be partially influenced by the animal's sex, as suggested by the significant differences observed in $Igf-I^{Ctrl/Ctrl}$ mice submitted to the MWM test which are not evident in $Igf-I^{\Delta/\Delta}$ mice. Furthermore, the conditional $Igf-I$ deletion in our mouse model eliminates the sexual dimorphic performance observed between control males and females in the OFT and rota-rod test, which could be in part because of the more

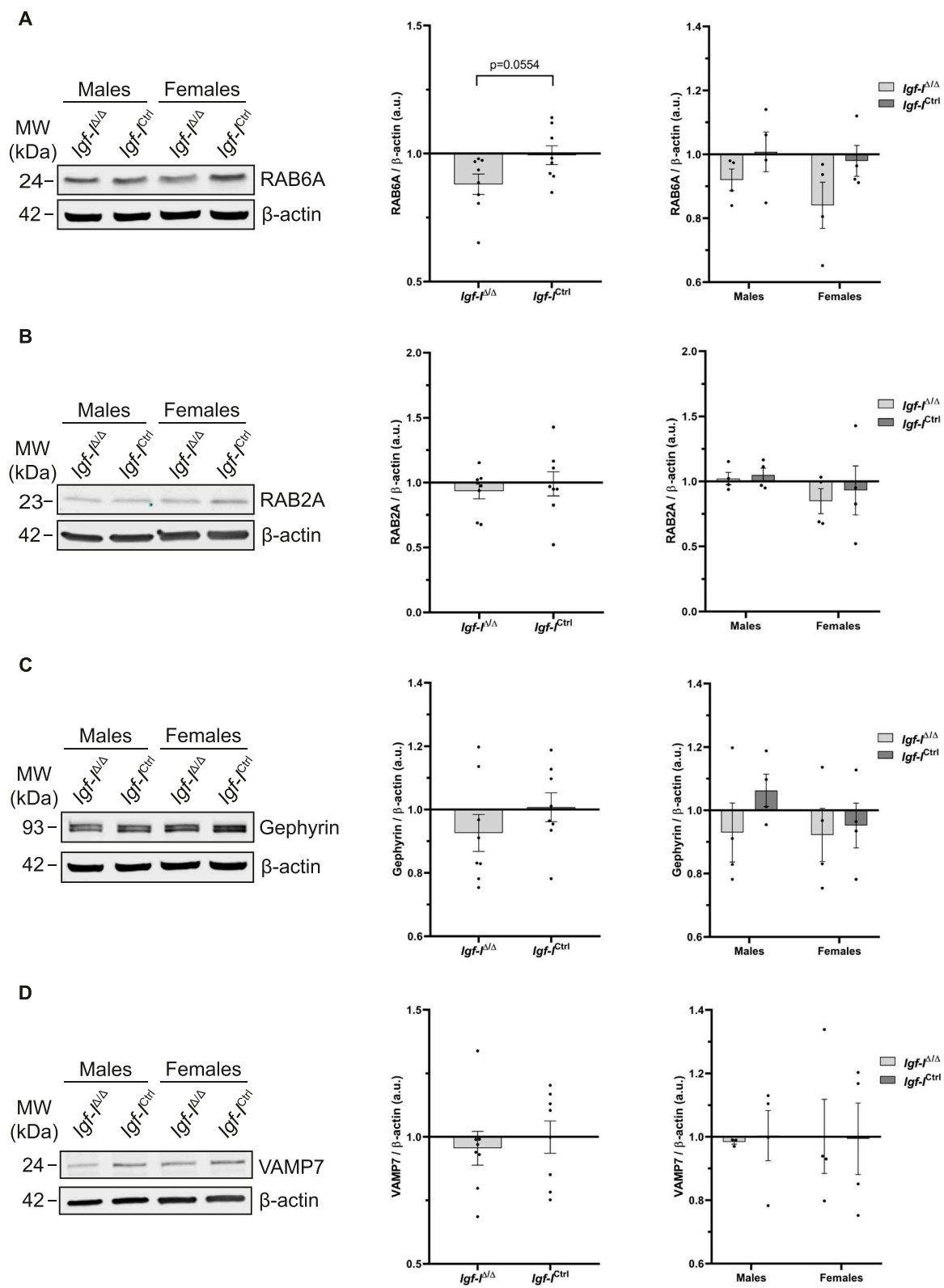

**Figure 9.  The effect of brain insulin-like growth factor-I (IGF-I) deletion on the expressions of RAB6A, RAB2A, gephyrin, and VAMP7.**
**(A, B, C, D)** Proteins extracted from the hippocampus of adult male and female *Igf-I*[Ctrl] or *Igf-I*[Δ/Δ] mice (6–16 mo-old) were separated by gel electrophoresis and transferred to membranes that were probed with specific antibodies against: (A) RAB6A, (B) RAB2A, (C) gephyrin, (D) VAMP7, and β-actin (A, B, C, D). Antibody binding was revealed with appropriate secondary antibodies and visualized using Odyssey CLx imaging system (see images on the left). The small decrease in the relative protein levels detected in *Igf-I*[Δ/Δ] versus *Igf-I*[Ctrl] mice were not significantly different (*t* test), although a clear trend was observed for RAB6A ($P = 0.0554$). The results are the mean ± SEM (n = 8 mice when data from males and females were combined) and (n = 4, when data from males and females were analyzed independently).

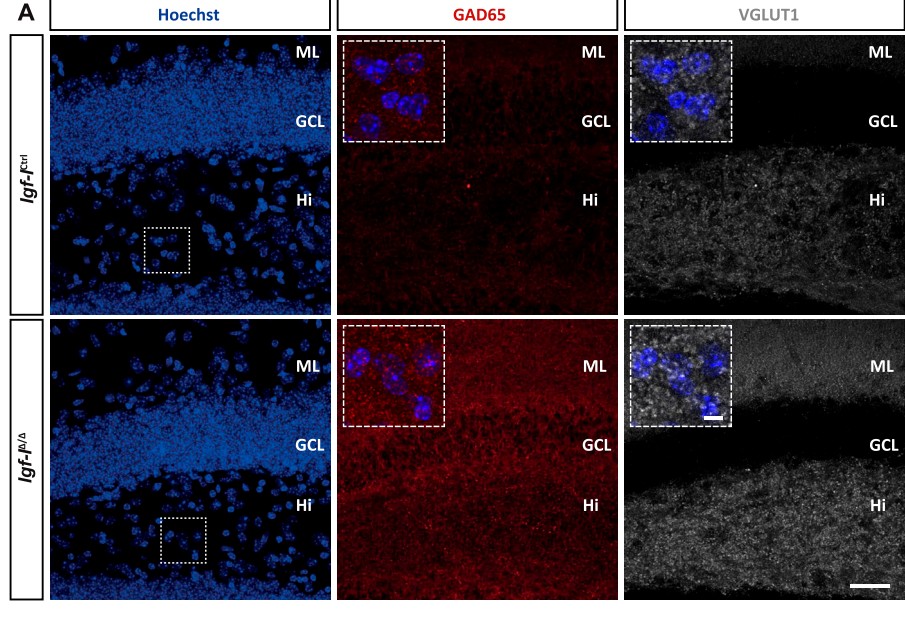

**Figure 10. The effect of brain insulin-like growth factor-I (IGF-I) deletion on the relative expression of GAD65 and VGLUT1.**
**(A)** Representative images of hippocampal sections immunostained with antibodies against GAD65 and VGLUT1, with the nuclei stained with Hoechst.
**(B, C)** The relative fluorescence units for GAD (B) and VGLUT1 (C) were 36% and 12% higher in the dentate gyrus of $Igf-I^{\Delta/\Delta}$ than in $Igf-I^{Ctrl/Ctrl}$ mice, respectively. The results are the mean ± SEM (n = 3 and six mice): *$P$ < 0.05. Scale bar = 30 $\mu m$; insets = 10 $\mu m$.

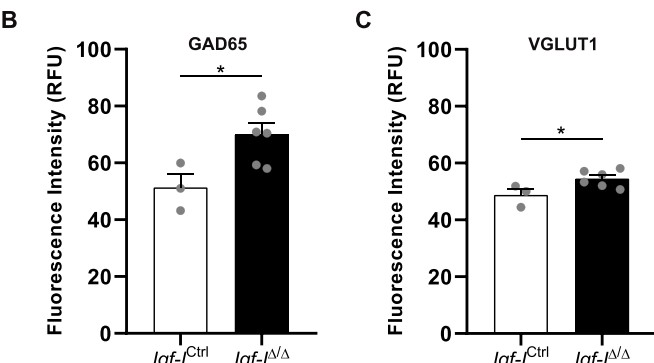

limited performance of $Igf-I^{\Delta/\Delta}$ female mice in terms of horizontal and vertical activities, stereotypy time, the time they spent in the central area, and the time they remained on the rota-rod. Therefore, brain-synthesized IGF-I is necessary to drive the sexual dimorphism in terms of the formation of new spatial memories, exploratory behavior, and motor learning and coordination.

Although our IHC assays revealed similar aromatase expression in the hippocampus of $Igf-I^{\Delta/\Delta}$ and $Igf-I^{Ctrl/Ctrl}$ mice, 25% of the hippocampal DEPs in female $Igf-I^{\Delta/\Delta}$ mice have been characterized as proteins that physically interact with estrogen receptors according to the BioGrid repository, whereas none of the hippocampal proteins altered in male $Igf-I^{\Delta/\Delta}$ mice are currently considered to interact with androgen receptors. Nonetheless, we found functional relationships between the DEPs in female and male $Igf-I^{\Delta/\Delta}$ mice with protein intermediates involved in estrogen and androgen metabolism, respectively. In addition, our analysis indicates the existence of functional relationships between proteins involved in neurogenesis and the altered hippocampal proteome of female but not of male $Igf-I^{\Delta/\Delta}$ mice.

The IGF-I signaling system has previously been implicated in the regulation of synaptic plasticity (Trejo et al, 2007; Fernandez & Torres-Aleman, 2012; Gazit et al, 2016; Pristera et al, 2019; Maglio et al, 2021; Williams et al, 2022) and in sexual dimorphic responses (Munive et al, 2016; Ashpole et al, 2017; Pinto-Benito et al, 2022), yet whether these effects were because of the autocrine/paracrine actions of this growth factor has been difficult to investigate. Using a genetic strategy, we present novel evidence that brain-synthesized IGF-I is crucial for hippocampal synaptic plasticity, and that it is involved in the regulation of spatial memory and sexual dimorphic behaviors that require exploratory activity and motor coordination. We identified alterations to the GCL organization, the inhibitory/excitatory ratio, and modules of synaptic proteins, which potentially underlie these processes. Because patients diagnosed with Alzheimer's disease show deficits in synaptic plasticity and in the acquisition of new memories (Wilson et al, 2023), our findings suggest that locally synthesized IGF-I is critical to maintain a healthy functional brain.

**Life Science Alliance**

**A**

**B**

**Figure 11. The effect of a brain insulin-like growth factor-I deletion on parvalbumin (PVA) neurons in the dentate gyrus.**
**(A)** Representative images of hippocampal sections immunostained with an anti-PVA antibody, with the nuclei stained with Hoechst. **(B)** Insulin-like growth factor-I deletion produced a nonsignificant increase in the number of PVA$^+$ neurons located in the granule cell layer, the molecular layer, and the Hilus (Hi). The results are the mean ± SEM (n = 3 and six mice) and the statistical analysis was performed using a $t$ test. Scale bar = 50 $\mu$m.

## Materials and Methods

### Ethical statement

All animal care and handling was carried out in accordance with European Union guidelines (directive 2010/63/EU) and Spanish legislation (Law 32/2007 and RD 53/2013). The protocols were approved by the Ethics Committees of the Consejo Superior de Investigaciones Científicas (CSIC, Madrid) and the Comunidad de Madrid (reference, PROEX 025/17). The animals were administered food and water ad libitum, and they were maintained under strictly controlled environmental conditions: 12-h light/dark cycle,

temperature 22°C, and humidity 44%. All efforts were made to ameliorate any animal suffering.

### Conditional *Igf-I* knock-out mice, Nestin-Cre:*Igf-I* (*Igf-I*$^{\Delta/\Delta}$)

The PCR genotyping and the general characteristics of the *Nestin-Cre*: *Igf-I* mice were described previously in detail (Nieto-Estevez et al, 2016b). In brief, the *Igf-I*$^{\Delta/\Delta}$ mice grow normally although their hippocampal DG and the GCL are significantly smaller and hippocampal neurogenesis is altered. In the present work, we analyzed 31 conditional *Igf-I* KO mice (*Igf-I*$^{\Delta/\Delta}$) and 32 control mice. The latter included *Igf-I*$^{+/+}$, *Igf-I*$^{fl/fl}$, *Igf-I*$^{+/fl}$, and *Igf-I*$^{+/+}$ Cre$^+$ mice. Statistically,

significant differences were not found between the different control mice (hereafter referred to as *Igf-I*[Ctrl]) and therefore, the results obtained from these genotypes were combined in the study. All the mice used in this study were on a C57Bl6N genetic background.

## Electrophysiological slice recordings

Hippocampal slices were obtained from *Igf-I*[Ctrl] and *Igf-I*[Δ/Δ] mice (14–15 mo old) after the animals had been anaesthetized and decapitated. To reduce the swelling and damage to the superficial layers of the slices from the adult/ageing mice, their brains were removed rapidly and placed in an ice-cold modified NMDG artificial cerebrospinal fluid (ACSF) gassed with 95% $O_2$/5% $CO_2$. This solution is used as a substitute for sodium ions over a wide range of adult ages and for different applications (Ting et al, 2014), and it contains (in mM): NMDG 93, KCl 2.5, $NaH_2PO_4$ 1.2, $NaHCO_3$ 30, HEPES 20, $MgSO_4$ 10, $CaCl_2$ 0.5, glucose 25, Sodium Ascorbate 5, thiourea 2, sodium pyruvate 3, (pH 7.3–7.4). Hippocampal slices (400 $\mu m$) were obtained in NMDG-ACSF and incubated for 10 min at 32°C before being transferred to regular ACSF at RT for 1 h before recording. This regular ACSF was also gassed with 95% $O_2$/5% $CO_2$ and it was made up of (in mM): NaCl 124, KCl 2.69, $KH_2PO_4$ 1.25, $MgSO_4$ 2, $NaHCO_3$ 26, $CaCl_2$ 2, glucose 10 (pH 7.3–7.4). The slices were then transferred to an immersion recording chamber and superfused with regular gassed ACSF. The hippocampus was visualized under an Olympus BX50WI microscope and extracellular recordings were obtained at 32°C. The stimulation and recording electrodes (2–4 MΩ, filled with NaCl 3 mM) were placed on the ML and GCL of the DG, respectively. Excitatory synaptic activity was isolated with the GABAa receptor antagonist picrotoxin (50 $\mu M$), and field excitatory postsynaptic potentialss were bandpass filtered between 0.3 Hz and 1.0 kHz, and digitized at 10.0 kHz. Ten-min baseline responses at 0.1 Hz were recorded before LTP protocol, which was achieved with 100 Hz trains delivered four times at 0.05 Hz. The pCLAMP 10 software (Molecular Devices) was used to generate the stimuli, and for the data display, acquisition, storage, and analysis.

Normality tests were performed before performing statistical comparisons. Then, one-way ANOVA followed by Tukey's post hoc test were applied for multiple comparisons.

## Behavioural tests

### MWM
To assess spatial learning and memory, animals (6–16 mo old) were placed in a water pool on successive days (Morris, 1984). On the pretraining day, the mice were submitted to a habituation session in the absence of a platform to determine if they have a preference for a particular quadrant of the pool and to rule out any motor problems. The acquisition phase was then carried out over four consecutive days during which the mice had to swim to reach a platform in the pool within a maximum time of 1 min, measuring the average escape latency (seconds) from the water to the platform in four trials each day. If the animal did not reach the platform, it was guided to the platform and it remained there for 30 s. Finally, in the transfer test on day 6, the animals were placed in the pool without the platform, and the time and distance they swam in the platform (P) quadrant or in the other three quadrants were measured using the *AnimalTracker* application (Gulyas et al, 2016) and ImageJ software (NIH, Bethesda, MD). The velocity and freezing time were also assessed. The tracking file obtained was examined with the *Tracking Analyzer module* and the data collected were analyzed statistically with GraphPad Prism 8 software (GraphPad).

### OFT
Mice (6–10 mo old) were introduced into a 42 × 42 × 30 cm arena (Versamax: AccuScan Instruments, Inc.) for 5 min, and the time spent exploring the center of the open field, and the total distance travelled, were quantified automatically with the manufacturer's software.

### Rota-rod test
Motor learning and coordination were assessed using a rota-rod apparatus (Ugo Basile) in six consecutive 5-min trials with constant acceleration. Animals were familiarized with the procedure under a constant rod speed the day before the tests started.

### Elevated plus maze
To assess anxiety-like/coping behavior, mice were introduced in a maze of 40 cm from the floor with two opposing, closed and open, arms. The test measures anxiety as a function of time spent in the open arms versus the closed arms. Time and entries in open and closed arms were scored, with more time spent in the open arms indicating less anxiety. Each animal was introduced into the middle of the apparatus for 5 min. Stress was scored as time spent in the closed arms, whereas coping behavior was estimated by time spent in the open arms. All measures were recorded (Video Tracking Plus Maze Mouse; Med Associates), and analyzed as described (Munive et al, 2019).

For statistical analysis, the data were analyzed with GraphPad Prism 6.0 or 8.0 software, first assessing a normal distribution of the data using a Kolmogorov–Smirnov test or a Shaphiro–Wilk test (for a small number of samples). The data were then analyzed with a *t* test with Welch correction for different variances, or one-way ANOVA followed by a Tukey post hoc test for equal variances, a Welch's one-way ANOVA test followed by Games–Howell post hoc test for different variances, or a two-way ANOVA followed by a Bonferroni or Sidak post hoc tests. $P < 0.05$ was considered significant and the ROUT method was applied to score potential data outliers.

## Sample preparation for hippocampal proteomics

Adult mice (10–15 mo old) were placed in a $CO_2$ atmosphere and then decapitated. After carefully removing the skull, the hippocampus was dissected out, frozen rapidly in liquid nitrogen and stored at –80°C until processing. ~0.5 mg of hippocampal tissue from the *Igf-I*[Ctrl] (or *Igf-I* control) or *Igf-I*[Δ/Δ] (or *Igf-I* conditional knockout -cKO- mice) was homogenized in lysis buffer containing 7 M urea, 2 M thiourea, and 50 mM DTT. The homogenates were centrifuged at 100,000$g$ for 1 h at 15°C. Protein precipitation was performed using the ReadyPrep 2-D Cleanup Kit (Bio-Rad) following

the manufacturer's instructions. Then, protein concentration was measured using the Bradford assay kit (Bio-Rad).

### Protein digestion, peptide TMT labelling, and off-line fractionation

Two independent TMT-based quantitative proteomic experiments were performed on three biological replicates from male and female $Igf$-$I^{Ctrl}$ and $Igf$-$I^{\Delta/\Delta}$ mice. TMT labelling of each sample was performed following the manufacturer's protocol (Thermo Fisher Scientific) and briefly, equal amounts of protein (600 μg) from each sample were reduced for 1 h at 55°C with 200 mM tris (2-carboxyethyl) phosphine. Cysteine residues were alkylated with 375 mM iodoacetamide at RT for 30 min and protein trypsin cleavage (1:40, w/w; Promega) was carried out at 37°C for 16 h. Peptide desalting was performed using Pierce Peptide Desalting Spin Columns according to the manufacturer's instructions. For TMT experiments, each tryptic digest was labelled with one isobaric amine-reactive tag as follows: (i) TMT-Plex-1 (male mice)—Tag126, $Igf$-$I$ control-1; Tag127, $Igf$-$I$ control-2; Tag128, $Igf$-$I$ control-3; Tag129, $Igf$-$I$ cKO-1; Tag130, $Igf$-$I$ cKO-2; Tag131, $Igf$-$I$ cKO-3; (ii) TMT-Plex-2 (female mice)—Tag126, $Igf$-$I$ control-1; Tag127, $Igf$-$I$ control-2; Tag128, $Igf$-$I$ control-3; Tag129, $Igf$-$I$ cKO-1; Tag130, $Igf$-$I$ cKO-2; Tag131, $Igf$-$I$ cKO-3. After 1 h of incubation at RT, the reactions were stopped by adding 5% hydroxylamine, and the labelled samples corresponding to the same plex were independently pooled, desalted, and evaporated in a vacuum centrifuge. The peptide pools were reconstituted with 40 μl of ammonium bicarbonate (ABC, 5 mM) at pH 9.8, and injected into an ÄKTA pure protein purification 25 system (GE Healthcare Life Sciences) with a high pH stable X-Terra RP18 column (C18, 2.1 × 150 mm, 3.5 μm; Waters). The mobile phases were 5 mM ammonium formate in 90% acetonitrile (ACN) at pH 9.8 (buffer B) and 5 mM ammonium formate in water at pH 9.8 (buffer A). The column gradient was established as an 80-min three-step gradient: from 5% B–30% B over 5 min, 30% B–60% B over 40 min, 15 min in 60% B, and 60% B–90% B over 20 min. The column was equilibrated in 95% B for 30 min and 2% B for 10 min. 12 fractions were collected and evaporated under vacuum.

### Mass spectrometry data acquisition

Before MS analysis, the peptide fractions were reconstituted to a final concentration of 0.5 μg/μl in 2% ACN, 0.5% formic acid (FA), and 97.5% ddH2O. The peptide mixtures were then separated by reverse phase chromatography using an Eksigent nanoLC ultra 2D pump fitted with a 75-μm ID column (0.075 × 250; Eksigent). Samples were first loaded for desalting and concentrating onto a 2 cm, 100 μm ID precolumn packed like the separating column. The mobile phases were 100% water/0.1% FA (buffer A) and 100% ACN/0.1% FA (buffer B). Non-modified peptide fractions were analyzed according under the following conditions. The column gradient was established as a 135-min three-step gradient from 2% B–30% B over 90 min, from 30% B–40% B over 10 min, and from 40–80% over 10 min. The column was equilibrated with 97% B for 3 min and 2% B for 23 min. Throughout the process, the precolumn was in line with the column and the flow was maintained along the gradient at 300 nl/min. The peptides eluted from the column were analyzed using a 5,600

Triple-TOF system (SCIEX) and the data were acquired upon a survey scan performed in a mass range from 350 up to 1,250 m/z, and with a scan time of 250 ms. LC-MS/MS control and data acquisition were performed using the Analyst TF1.7 software (SCIEX). The top 35 peaks were selected for fragmentation and a minimum accumulation time for MS/MS was set at 11 ms, giving a total cycle time of 3.8 s. The product ions were scanned in a mass range from 100 m/z up to 1,500 m/z and excluded for further fragmentation for 15 s.

### Proteomic data analysis

The raw MS/MS spectra were processed using the MaxQuant v1.6.17 software (Tyanova et al, 2016a) and used to search the Uniprot proteome reference for mouse. The parameters employed were: initial maximum precursor (25 ppm); fragment mass deviations (40 ppm); variable modification (methionine oxidation and N-terminal acetylation), and fixed modification (MMTS); enzyme (trypsin) with a maximum of 1 missed cleavage; minimum peptide length (seven amino acids); false discovery rate for peptide-spectrum match and protein identification (1%). Frequently observed laboratory contaminants were removed. Protein identification was considered valid with at least one unique or "razor" peptide. Protein quantification was calculated using at least two razor plus unique peptides, and statistical significance was calculated with a two-way $t$ test ($P < 0.05$), with a 1.3-fold change cut-off. Proteins with TMT ratios below the low range threshold (0.77) were considered to be down-regulated, whereas those above the high range threshold (1.3) were considered to be up-regulated. The Perseus v1.6.14.0 software (Tyanova et al, 2016b) was used for statistical analysis and data visualization. The pathway and interactome analyses were performed using the BioGrid 4.4 (Oughtred et al, 2021), STRING 11.5 (Szklarczyk et al, 2019), Ingenuity (QIAGEN), and Metascape tools (Zhou et al, 2019).

### Tissue collection and Western blotting

Animals (6–16 mo old) previously used in the MWM test were euthanized and their brains were removed from the skulls. The hippocampus was isolated using a magnifying glass and homogenized mechanically in RIPA buffer (R0278; Sigma-Aldrich, Merck) supplemented with protease and phosphatase inhibitor cocktails (Roche, Merck). The extracts were then centrifuged at 16,000$g$ at 4°C for 10 min, collecting the supernatants and storing them at −80°C until further use. The concentration of the proteins obtained was determined by the BCA method (BCA Protein assay kit, 23227; Thermo Fisher Scientific). Hippocampal samples were then diluted in Laemmli buffer (60 mM Tris–HCl [pH 6.8], 2% SDS, 10% glycerol, 5% ß-mercaptoethanol, and 0.01% bromophenol blue), boiled for 5 min at 100°C, and resolved on precast 4–20% SDS-polyacrylamide gels (Mini-Protean TGX gels, 4561094; Bio-Rad). The proteins were transferred to a 0.2-μm nitrocellulose membrane (Amersham Protran nitrocellulose blotting membrane, 10600094; Merck) and the membrane was blocked for 1 h at RT with 5% nonfat dry milk in TBS plus Tween 20 (20 mM Tris–HCl [pH 7.6], 140 mM NaCl and 1% Tween 20). The membrane was probed with the primary

antibodies overnight at 4°C: anti-RAB6A (1:2,000: NBP1-33110; Novus biologicals), anti-RAB2A (1:2,000: A95802; Antibodies.com), anti-Gephyrin (1:2,000: PA5-29036; Invitrogen, Thermo Fisher Scientific), anti-VAMP7 (1:1,000: 232003; Synaptic Systems), and anti-$\beta$-actin (1: 10,000: A2228; Sigma-Aldrich, Merck). Antibody binding was detected with IRDye 800CW and 680RD secondary antibodies (1:20,000; LI-COR Biosciences) for 1 h at RT and the images were acquired with the Odyssey CLx imaging system (I-I-COR Biosciences). Densitometric quantification was performed using Image Studio Lite acquisition software (LI-COR Biosciences), subtracting the background and normalizing the signal was to that of ß-actin as an internal control. Each sample was assayed in triplicate and the statistical analysis was performed using a *t* test to compare delta/delta with Ctrl/Ctrl, and with a two-way ANOVA and Sidak post hoc test to include the sex variable, all performed using Graph Pad 8.0. The ROUT method was applied previously to determine potential outliers.

### Tissue collection and immunohistochemistry

Adult mice (12 mo old) were anaesthetized with an intraperitoneal (ip) pentobarbital and then perfused transcardially with 0.9% NaCl followed by 4% PFA in PB. The animal's brain was post-fixed for 48 h, embedded in 3% agarose, and 40 $\mu$m vibratome serial sections were processed for immunohistochemistry. Sections were permeabilized in a solution containing 0.4% Triton X-100 and 10% normal donkey serum, and then incubated overnight at RT with a primary antibody against: aromatase (kindly provided by Dr Pablo Méndez, Instituto Cajal, Madrid, Spain); GAD65 (1:50: Cat# MAB351, RRID:AB_2263126; Millipore); gephyrin (1:500: Cat# PA5-29036, RRID: AB_2546512; Invitrogen/Thermo Fisher Scientific); MAP2 (1:200–1: 500: Cat# M1406, RRID:AB_477171; Sigma-Aldrich); Prox1 (1:750: Cat# AB5475, RRID:AB_177485; Millipore); PSD95 (1:500: Cat# 18258, RRID: AB_444362; Abcam); PVA (1:750: Cat# 235, RRID:AB_10000343; Swant); RAB6A (1:250: Cat# NBP1-33110, RRID:AB_2175468; Novus biologicals), Synapsin I (1:500: Cat# AB1543, RRID:AB_ 2200400; Millipore); VGLUT1 (1:500: Cat# AB5905, RRID:AB_2301751; Millipore). Subsequently, the samples were washed with PBS and incubated for 4 h at RT in the dark with the appropriate Alexa Fluor secondary antibodies. The sections were then washed, exposed to Hoechst (1 $\mu$g/ml: Cat# H3570; Thermo Fisher Scientific), and mounted on glass microscope slides in Mowiol. No specific signal was detected in the absence of the primary antibodies but in the presence of the secondary antibodies.

### Cell counting and analysis of the relative fluorescence intensity (RFU)

Neurons located in the DG and that express aromatase, MAP2, Prox1, PVA, and RAB6A were counted in confocal images taken from serial sections. Individual z-planes were collected at a resolution of 1,024 × 1,024 every 1–2 $\mu$m with a 40x objective. The entire Z-stack was counted to calculate the number of immunoreactive cells, and to assess the co-localization between MAP2 and RAB6A and between MAP2 and aromatase. Co-localization was confirmed by analyzing each individual plane of each section using ImageJ software. The results are shown as the mean (±SEM) of MAP2⁺/ RAB6A⁺ neurons and of MAP2⁺/aromatase⁺ neurons relative to the

total number of MAP2⁺ neurons or as the mean (±SEM) of the labelled cells per volume as determined using the ImageJ software on confocal Z-stacks. For all the IHC experiments, two sections per animal from three *Igf-I^Ctrl* and five to six *Igf-I^{Δ/Δ}* animals were analyzed per condition ("n" value = 3–6).

The fluorescence intensity (given as the RFU) of the orthogonal projection images of the DG was measured using Fiji-ImageJ2 after subtracting the intensity from the negative controls (i.e., sections incubated with secondary antibodies alone). Briefly, individual z-planes at a resolution of 1,024 × 1,024 were taken every 0.5–1.0 $\mu$m with a 40x objective and the maximal orthogonal projection per z-stack was obtained. After establishing the threshold from the images from negative controls, the average fluorescence intensity was measured and annotated. This value was subtracted from each projection image of the sections incubated with both primary and secondary antibodies. The threshold was established, selecting pixels with specific labelling and measuring the average fluorescence intensity. Blood vessels and autofluorescent aggregates were excluded from the analysis.

An unpaired two-tailed *t* test was used to compare the mean ± SEM values from *Igf-I^{Δ/Δ}* and *Igf-I^Ctrl* mice, with a Welch's correction when the F-test indicated significant differences between the variances of both groups.

## Data Availability

The files of the MS data and search results were deposited in the Proteome Xchange Consortium via the JPOST partner repository (https://repository.jpostdb.org—[Okuda et al, 2017]), with the identifier PXD028458 for ProteomeXchange and JPST001316 for jPOST.

## Supplementary Information

## Acknowledgements

We thank Lucía Vicario (Instituto Cajal-CSIC, Madrid, Spain) for helping with the composition of the figures, and Dr. M Sefton (BiomedRed SL, Madrid, Spain) for English editing. This work was funded by grants from the Spanish "Ministerio Ciencia, Innovación y Universidades, and the Ministerio de Ciencia e Innovación/Agencia Estatal de Investigación" (MICIU and MICINN/AEI SAF2016-80419-R, PID2019-109059RB-100, and CIBERNED CB06/ 05/0065 to C Vicario; PID2019-110356RB-I00/AEI/10.13039/501100011033 to J Fernández-Irigoyen and E Santamaría; PID2019-104376RB-I00 to I Torres-Alemán; and PID2019-106579RB-I00 funded by MCIN/AEI/10.13039/ 501100011033 and by "ERDF A way of making Europe to G Perea, and BES-2017-080303 to C González-Arias).

### Author Contributions

R Herrero-Labrador: formal analysis, investigation, and methodology.
J Fernández-Irigoyen: data curation, formal analysis, supervision, investigation, and methodology.

R Vecino: software, formal analysis, validation, investigation, visualization, methodology, and writing—review and editing.

C González-Arias: formal analysis, investigation, and methodology.

K Ausín: investigation and methodology.

I Crespo: investigation and methodology.

FJ Fernández Acosta: formal analysis, validation, investigation, visualization, and writing—review and editing.

V Nieto-Estévez: methodology.

MJ Román: investigation and methodology.

G Perea: conceptualization, resources, formal analysis, supervision, funding acquisition, methodology, and writing—original draft and project administration.

I Torres-Alemán: conceptualization, resources, formal analysis, supervision, funding acquisition, methodology, and writing—original draft and project administration.

E Santamaría: conceptualization, resources, data curation, formal analysis, funding acquisition, methodology, and writing—original draft and Project administration.

C Vicario: conceptualization, resources, formal analysis, supervision, funding acquisition, validation, investigation, visualization, methodology, and writing—original draft and project administration.

## Conflict of Interest Statement

The authors declare that they have no conflict of interest.

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
