## [Reviewer comments · Life Science Alliance]

Life Science Alliance

Brain IGF-I regulates LTP, spatial memory and sexual dimorphic behavior

Carlos Vicario, Raquel Herrero-Labrador, Joaquín Fernández-Irigoyen, Rebeca Vecino, Candela González-Arias, Karina Ausín, Inmaculada Crespo, Francisco Fernández Acosta, Vanesa Nieto-Estévez, María Román, Gertrudis Perea, Ignacio Torres-Alemán, and Enrique Santamaría

DOI: <https://doi.org/10.26508/lsa.202201691>

Corresponding author(s): Carlos Vicario, Spanish National Research Council and Enrique Santamaría,

Review Timeline:

Submission Date:	2022-08-25
Editorial Decision:	2022-09-29
Revision Received:	2023-05-30
Editorial Decision:	2023-06-26
Revision Received:	2023-06-29
Accepted:	2023-07-03

Scientific Editor: Novella Guidi

Transaction Report:

September 29, 2022

Re: Life Science Alliance manuscript #LSA-2022-01691

Dr. Carlos Vicario-Abejón
Spanish National Research Council
Molecular, Cellular biology and development
Avenida Dr. Arce 37
Madrid, Madrid 28002
Spain

Dear Dr. Vicario-Abejón,

Thank you for submitting your manuscript entitled "Brain IGF-I is essential for LTP, and it regulates sexual dimorphic behaviour and proteostasis" to Life Science Alliance. The manuscript was assessed by expert reviewers, whose comments are appended to this letter. We invite you to submit a revised manuscript addressing the Reviewer comments.

Thank you for this interesting contribution to Life Science Alliance. We are looking forward to receiving your revised manuscript.

Sincerely,

B. MANUSCRIPT ORGANIZATION AND FORMATTING:

Reviewer #1 (Comments to the Authors (Required)):

The manuscript presented by Herrero-Labrador et al shows how Igf-1 deletion significantly changes the hippocampal proteome on male and female mice and this conclusion was drawn after a significant reduction or ablation of LTP.

The work is sound and of relevance with minor and major issues to be solved:

Minor issues:

- there are small typos throughout the manuscript, please correct
- in the text it is mentioned " $p < 0.05$ " while in some figures it is stated " $p \leq 0.05$ ". Please confirm in the entire manuscript which one was used and correct accordingly. In some cases, there is repetition of this information in the text and in the figures captions. It is redundant. Please avoid it.
- on page 6, "(Figure 2A,B)" is mentioned on the second sentence of the paragraph while the first sentence is already about figure 2. Please indicate each figure as soon as possible in the sentences.
- Figure 3 is quite short and I believe that it could have been merged with figure 2, making the necessary adjustments to the figure title and captions.
- page 7 "No proteins were deregulated uniformly across both cohorts." Is repeated with the information inside the figure caption.
- avoid "data not shown" as it cannot be peer-reviewed. Therefore, any sentence about missing data can not be evaluated. If needed, add this data as supplementary material.
- on the ethics section, please provide the authorization code for this project.
- explain the protein precipitation step in the sample preparation for proteomics
- MilliQ water is a nonscientific term, and it refers to a brand. Authors are most likely referring to double deionized water (ddH₂O).
- please follow the MIAPE guidelines with special care on indicating all software used in all steps (from data acquisition to data analysis) including their versions. P.e. Analyst is missing.
- The IHC section is indicated but there are no results based on this sample prep.
- Figure 1 Caption, please indicate that "*" refers to the analysis in relation to the base line while "#" refers to the analysis between the two experimental groups.
- Figure 2, is it "<" or " \leq "? There is no text for two asterisks in the caption. Use just a straight line below the asterisk and center the beginning and end of the line with the center of the respective scatter data.
- Figure 4 doesn't have a title. For quicker analysis, please provide in supplementary material the same image but with the protein names aligned from both genders, to quickly observe that there are no differences in the trends of the same protein in both sexes.
- Supplementary figure 1, please indicate the test which was used and retrieved no statistical differences.

Major issues:

- could the authors clarify how the proteomics experiments were performed and why there are around 2000 proteins quantified in the female dataset and around 3300 proteins quantified in the male dataset? How was the normalization performed considering such a huge difference in the number of quantified proteins?
- Figure 5 is quite complex to analyze and in some sense difficult to retrieve meaningful information. The authors could try to use other tools which allow a much quicker comparison of number of genes being regulated such as the one presented by Bonnot et al (DOI:10.21769/BioProtoc.3429)
- validation of the results is very important and there are conclusions being drawn linking the observed phenotype and the proteomics changes. However, the electrophysiology experiments were performed in neurons and the proteome was performed in tissue, which comprises different cell types. It is important to understand if the proteins being modulated are in neurons or other cell type. In this case, IHC experiments on some of the most relevant proteins should be performed along with neuronal markers.

Reviewer #2 (Comments to the Authors (Required)):

In their manuscript, Herrero-Labrador et al., investigate the effect of brain IGF-1 on neuronal activity, behavior and protein homeostasis. Herrero-Labrador and colleagues use the Nes-Cre mouse to delete IGF-1 in the brain and investigate long-term synaptic plasticity by recording excitatory postsynaptic field potentials from hippocampal slices. Here, brain slices of KO animals do not exhibit synaptic potentiation. Further they investigate behavior and show that IGF-1 is an important factor for sex-specific behavior in the open field test. Lastly they analyze the proteome of hippocampal samples of control and KO animals. This analysis revealed that IGF-1 modulates the hippocampal proteome, which altered protein expression related to "vesicle-related compartments" and potentially also a link to sex hormone signaling.

While reading the manuscript, I recognized that all mouse experiments have been conducted at different ages. As age is an important confounder, it is therefore difficult to link alterations in neuronal activity (14-15 months) and protein homeostasis (10-15 months) to the observed differences in behavior (6-10 months). Unfortunately, the different parts of the manuscript are not well interconnected and rather seem as a line of different observations. While hippocampal slices are used to assess neuronal activity and protein homeostasis, the behavioral tests depend on motor function and exploration. Thus a classical memory test such as the Water maze test at the correct age should have been used to link molecular alterations in the hippocampus to the behavior.

I had the following issues with their presented data and conclusions in this manuscript.

- 1: Hippocampal-dependent behavior needs to be investigated at the same age, as neuronal activity and the proteome analysis has been performed.
2. The authors state that there is no differences between groups in the Barnes maze test (Suppl. Fig.1). Yet only three female control mice have been investigated. This number is far too small to make hard conclusions. Overall the n is small for behavioral assessment (except male controls).
- 3: Important altered proteins (Fig. 5-7) should have been confirmed by western blotting and tested whether they are regulated by sex hormones (in vitro).
- 4: Does brain IGF-1 differ between male and females in these animals?
4. Overall the figures are not precise enough described (especially Fig.5) making it very difficult to follow and understand reasons for the analysis.
- 5: In the methods part "tissue collection and immunohistochemistry" has been described, e.g. Hoechst and Prox1 staining, but no data have been shown. What is the reason for this?

Minor:

There are several grammar mistakes in the manuscript.

At which specific dates have the behavioral tests been conducted?

The genetic background should be stated, as the authors can not ask all readers to read their first manuscript (Nieto-Estevez et al 2016b). Moreover the citations should be clearer defined as e.g. Nieto-Estevez et al 2016a and b has not been listed as 2016 a and b.

Manuscript # LSA-2022-01691, now titled “Brain IGF-I regulates LTP, spatial memory and sexual dimorphic behavior”

Point by point responses to the Reviewer’s comments.

We would like to express our thanks to the Reviewers for their constructive comments and criticisms, which have allowed us to more clearly transmit the main message of our paper, i.e. that brain synthesized IGF-I is critical for hippocampal LTP, spatial memory formation and the regulation of sexual dimorphic behavior. Furthermore, we show that brain IGF-I is necessary to maintain the structure of the granule cell layer, and that it regulates the inhibitory/excitatory balance and protein network modules that possibly have an impact on synaptic plasticity and memory.

Please, note that the major changes made in the manuscript are written in blue.

Reviewer #1

Thank you for your comments on our manuscript, please find our responses to all the issues you raised below.

Major issues:

(1) Could the authors clarify how the proteomics experiments were performed and why there are around 2000 proteins quantified in the female dataset and around 3300 proteins quantified in the male dataset? How was the normalization performed considering such a huge difference in the number of quantified proteins?

A unique normalization step encompassing all hippocampal samples was not possible mainly due to the different timing in the processing of both experimental groups. In that time, several changes/improvements to the mass-spectrometry phase were performed, obtaining more quantitative data in the female dataset, which led us to establish two independent processes of normalization. However, it is important to point out that male and female mice were age-matched when their brains were extracted.

(2) Figure 5 is quite complex to analyze and in some sense difficult to retrieve meaningful information. The authors could try to use other tools which allow a much quicker comparison of number of genes being regulated such as the one presented by Bonnot et al (DOI:10.21769/BioProtoc.3429)

Based on this problem we decided to move Fig. 5 to the supplementary material (Supplementary Fig. S4), not least as we feel that the main message of this figure is just as clearly shown in Fig. 5B (a common alteration to membrane trafficking in males and females).

(3) Validation of the results is very important and there are conclusions being drawn linking the observed phenotype and the proteomics changes. However, the electrophysiology experiments were performed in neurons and the proteome was performed in tissue, which comprises different cell types. It is important to understand if

the proteins being modulated are in neurons or other cell type. In this case, IHC experiments on some of the most relevant proteins should be performed along with neuronal markers.

Thank you for pointing this out and to address this, we have first used a specific antibody to Prox1, a marker of granule neurons (which in the previous version was mentioned as data not shown). The results obtained indicate that brain *Igf-I* deletion produces significant increases in the number of ectopic Prox1⁺ neurons outside the granule cell layer (GCL) and specifically, in the molecular layer (ML) and the hilus (Hi) (Fig. 8). We suggest that this alteration may have a negative impact on the induction of LTP.

Next, we performed IHC experiments using a battery of antibodies against relevant proteins (RAB6A and gephyrin), proteins identified in the proteomic analysis, although we were unable to find a suitable antibody for IHC to study others (e.g., RAB2A). In addition, and as mentioned in the manuscript, interfacing the differentially expressed proteomes with the SYNGO and Ingenuity Pathway Analysis repositories revealed alterations to synaptic proteins and to proteins involved in LTP. Consequently, we also carried out IHC to detect presynaptic (synapsin I) and postsynaptic proteins (PSD95), as well as proteins with a relevant role in regulating the activity of inhibitory (GAD65 and PVA) and excitatory (VGLUT1) synaptic circuits. Moreover, since proteins involved in estrogen and androgen metabolism were functionally connected to some of the deregulated hippocampal proteins, we performed IHC to label aromatase (an enzyme catalyzing estrogen synthesis from androgens). These results are now presented in the new Figures 10 and 11, and in the Supplementary Figures S6, S7, S8 and S9, and they lead us to draw two conclusions: 1) that brain *Igf-I* deletion could produce an imbalance in the inhibitory/excitatory ratio that affects LTP, and 2) that alterations to LTP and to synaptic protein modules may underlie the changes observed in synaptic plasticity and spatial memory.

Minor issues:

(1) There are small typos throughout the manuscript, please correct

We apologize for this. The manuscript has now been carefully revised throughout.

(2) In the text it is mentioned " $p < 0.05$ " while in some figures it is stated " $p \leq 0.05$ ". Please confirm in the entire manuscript which one was used and correct according. In some cases, there is repetition of this information in the text and in the figures captions. It is redundant. Please avoid it.

We apologize for this inconsistency. We confirm that $P < 0.05$, $P < 0.01$, $P < 0.001$ and $P < 0.0001$ has been used in the statistical analysis.

(3) On page 6, "(Figure 2A,B)" is mentioned on the second sentence of the paragraph while the first sentence is already about figure 2. Please indicate each figure as soon as possible in the sentences.

This has now been modified accordingly. Thank you.

(4) Figure 3 is quite short and I believe that it could have been merged with figure 2, making the necessary adjustments to the figure title and captions.

The original figures 2 and 3 have now been merged into the new figure 4. Thank you for this suggestion.

(5) Page 7 "No proteins were deregulated uniformly across both cohorts." Is repeated with the information inside the figure caption.

I'm sorry, but we do not feel this idea is repeated in the figure caption.

(6) Avoid "data not shown" as it cannot be peer-reviewed. Therefore, any sentence about missing data cannot be evaluated. If needed, add this data as supplementary material.

As mentioned above, we have used a specific antibody to Prox1, a marker of granule neurons, data that was referred to as "not shown" in the previous version of the manuscript. The results indicate that brain *Igf-I* deletion produces a significant increase in the number of ectopic Prox1⁺ neurons located outside the granule cell layer (GCL), specifically, in the molecular layer (ML) and the hilus (Hi).

(7) On the ethics section, please provide the authorization code for this project.

The authorization code for this project is now provided in the first paragraph of the Materials and Methods.

(8) Explain the protein precipitation step in the sample preparation for proteomics

According to the reviewer's suggestion, the following sentence has been added to the Materials and Methods section: "Protein precipitation was performed using the ReadyPrep™ 2-D Cleanup Kit (Bio-Rad) following the manufacturer's instructions".

(9) MilliQ water is a nonscientific term, and it refers to a brand. Authors are most likely referring to double deionized water (ddH₂O).

We have now modified this accordingly.

(10) Please follow the MIAPE guidelines with special care on indicating all software used in all steps (from data acquisition to data analysis) including their versions. P.e. Analyst is missing.

Thank you for this suggestion and for highlighting the oversight. The Minimum Information About a Proteomics Experiment (MIAPE) is currently employed in the new version of the manuscript, indicating the Software tools and the versions used (Analyst, Maxquant, Perseus, Biogrid, String).

(11) The IHC section is indicated but there are no results based on this sample prep.

The IHC section has now been completed and the new results obtained are shown in Figures 8, 10 and 11, and in the Supplementary Figures S6-S9.

(12) Figure 1 Caption, please indicate that "*" refers to the analysis in relation to the base line while "#" refers to the analysis between the two experimental groups.

The meaning of the symbols has now been clarified in the legend to Figure 1. Thank you for pointing out this oversight.

(13) Figure 2, is it "<" or "{less than or equal to}"? There is no text for two asterisks in the caption. Use just a straight line below the asterisk and center the beginning and end of the line with the center of the respective scatter data.

We apologise for this inconsistency, it is *P<0.05 and **P<0.01, as now indicated in the legend to Figure 4. The figure has now been modified as suggested by the reviewer.

(14) Figure 4 doesn't have a title. For quicker analysis, please provide in supplementary material de same image but with the protein names aligned from both genders, to quickly observe that there are no differences in the trends of the same protein in both sexes.

A title has been incorporated into Figure 4 (now Figure 5). Due to the experimental set-up, it was not possible to process hippocampal samples from male and female mice at the same time, although the animals were grouped by the age at which their brains were extracted (age-matched). During both independent analyses the mass spectrometer was cleaned, upgraded and several replacement parts were added, with the ensuing effect on performance. This is the main reason why it is not possible to produce a global heat-map representing all the proteomes quantified in male and female animals.

(15) Supplementary figure 1, please indicate the test which was used and retrieved no statistical differences.

The previous Supplementary figure 1 has been modified to include only the results obtained with the elevated plus maze (please see the response to Reviewer #2 below). Nevertheless, this data was analyzed using two-way ANOVA followed by a Bonferroni's post hoc test.

Reviewer #2

Thank you for your constructive comments on our manuscript and please find our responses to all the issues you raised below.

(1) Hippocampal-dependent behavior needs to be investigated at the same age, as neuronal activity and the proteome analysis has been performed.

Following the reviewer's comment, the performance of new mice was assessed in the Morris water maze (MWM) to address whether spatial learning and memory were altered by the lack of brain IGF-I. These mice were 6-16 months old, an age interval covering the ages of the mice used previously to assess neuronal activity (14-15 months), behavior (6-10 months) and protein homeostasis (10-15 months). The results obtained from the MWM assay indicated that *Igf-I^{Δ/Δ}* mice had higher escape latencies, spent significantly less time and swam significantly shorter distances in the platform quadrant (P) than *Igf-I^{Ctrl}* mice, further evidence that brain IGF-I plays a crucial role in the formation of new spatial memories. Moreover, our results indicate that the effect of brain IGF-I on spatial memory formation is partially sex-dependent. These new and important results are presented in Figures 2, 3 and Supplementary Figure S1, S2.

(2) The authors state that there is no differences between groups in the barnes maze test (Suppl. Fig.1). Yet only three female control mice have been investigated. This number is far too small to make hard conclusions. Overall the n is small for behavioral assessment (except male controls).

We agree with the Reviewer that using only three female control mice is too small a number to reach a clear conclusion. However, as indicated above we have now analyzed new animals (a total of 9 *Igf-I^{Ctrl}* and 9 *Igf-I^{Δ/Δ}* mice) using the MWM maze, which gave us more relevant results. As such, we have now removed the Barnes maze data from the manuscript.

(3) Important altered proteins (Fig. 5-7) should have been confirmed by western blotting and tested whether they are regulated by sex hormones (in vitro).

Thank you for pointing this out and as a result, we have now probed western blots with a battery of antibodies against relevant proteins identified in the proteomic analysis: RAB6A, RAB2A, Gephyrin, and VAMP7. Although the relative protein levels showed a tendency to decrease in *Igf-I^{Δ/Δ}* mice, these changes did not reach statistical significance ($P < 0.05$, P value for RAB6A was 0.0554; Figure 9, Supplementary Figure S5). However, unfortunately we were unable to address whether these proteins are regulated by sex hormones *in vitro*. Nevertheless, we did perform IHC experiments using antibodies against RAB6A and gephyrin, as mentioned above in response to Reviewer#1, yet for other proteins we could not find a suitable antibody for IHC (e.g., RAB2A). In addition, and as mentioned in the manuscript, associating the differentially expressed proteomes with the SYNGO and Ingenuity Pathway Analysis repositories revealed alterations to synaptic proteins and proteins involved in LTP. Hence, we also carried out IHC to evaluate presynaptic (synapsin I) and postsynaptic proteins (PSD95), as well as proteins that play a relevant role in regulating the activity of inhibitory (GAD65 and PVA) and excitatory (VGLUT1) synaptic circuits. Moreover, since proteins involved in estrogen and androgen metabolism were functionally connected with some of the deregulated hippocampal proteins, we performed IHC to label aromatase (an enzyme catalyzing estrogen synthesis from androgens). These results are now presented in Figures 10 and 11, and in Supplementary Figures S6, S7, S8 and S9, and they lead us to conclude 1) that brain *Igf-I* deletion could produce an imbalance in the inhibitory/excitatory ratio that affects

LTP, and 2) that alterations to LTP and to synaptic protein modules may underlie the changes observed in synaptic plasticity and spatial memory.

(4) Does brain IGF-1 differ between male and females in these animals?

We previously, analyzed the macroscopic phenotypes of the Nestin-Cre:Igf-I mice (Nieto-Estévez et al., 2016b) and we found that the *Igf-I^{Δ/Δ}* mice had a similar body weight to the *Igf-I^{Ctrl}* mice. We did not find differences in the volumes of any brain area studied, except that of the OB and the GCL of the DG, which were significantly smaller in the *Igf-I^{Δ/Δ}* mice. Potential differences between male and female brains were not studied nor was the expression of IGF-1.

(5) Overall the figures are not precise enough described (especially Fig.5) making it very difficult to follow and understand reasons for the analysis.

We apologize for this. We believe that the data in the figures are now better described and the reasons for the analysis are more precisely defined in the new version of the manuscript. Following the comments of both reviewers, we decided to move the original Fig. 5 to the supplementary material (Fig. S4). The main message transmitted by this figure is that the alteration to membrane trafficking is common to males and females, and we consider that this is just as clearly presented in Fig. 4B.

(6) In the methods part "tissue collection and immunohistochemistry" has been described, e.g. Hoechst and Prox1 staining, but no data have been shown. What is the reason for this?

We apologise for this inconsistency. In the revised version of the manuscript we have used a specific antibody raised against Prox1, a marker of granule neurons (which in the previous version was mentioned as data not shown). The results indicate that brain *Igf-I* deletion produces a significant increase in the number of ectopic Prox1⁺ neurons located outside the granule cell layer (GCL), and specifically in the molecular layer (ML) and hilus (Hi). We suggest that this alteration could have a negative impact on LTP induction.

Minor:

(1) There are several grammar mistakes in the manuscript.

We apologize for this and the manuscript has now been carefully edited to correct any grammatical errors.

(2) At which specific dates have the behavioral tested been conducted?

The behavioral tests included in the first version of the manuscript were carried out between April 2017 and April 2018. The water maze test, and the data acquisition and analysis, was carried out from October 2022.

3. The genetic background should be stated, as the authors cannot ask all readers to read their first manuscript (Nieto-Estevez et al 2016b). Moreover, the citations should be clearer defined as e.g. Nieto-Estevez et al 2016a and b has not been listed as 2016 a and b.

The reviewer is correct and we now indicate that all the mice used in this study were on a C57Bl6N genetic background. Moreover, the citations are now clearer defined:

- Nieto-Estevez V, Defterali C, Vicario-Abejon C (2016a) IGF-I: A Key Growth Factor that Regulates Neurogenesis and Synaptogenesis from Embryonic to Adult Stages of the Brain. *Front Neurosci* 10:52. doi:10.3389/fnins.2016.00052

-Nieto-Estevez V, Oueslati-Morales CO, Li L, Pickel J, Morales AV, Vicario-Abejon C (2016b) Brain Insulin-Like Growth Factor-I Directs the Transition from Stem Cells to Mature Neurons During Postnatal/Adult Hippocampal Neurogenesis. *Stem Cells* 34 (8):2194-2209. doi:10.1002/stem.2397

June 26, 2023

RE: Life Science Alliance Manuscript #LSA-2022-01691R

Dr. Carlos Vicario-Abejón
Spanish National Research Council
Molecular, Cellular biology and development
Avenida Dr. Arce 37
Madrid, Madrid 28002
Spain

Dear Dr. Vicario-Abejón,

Thank you for submitting your revised manuscript entitled "Brain IGF-I regulates LTP, spatial memory and sexual dimorphic behavior". We would be happy to publish your paper in Life Science Alliance pending final revisions necessary to meet our formatting guidelines.

- please address the final Reviewer 2's concerns
- please upload your main manuscript text as an editable doc file;
- please add ORCID ID for the corresponding secondary author--they should have received instructions on how to do so
- please add the Twitter handle of your host institute/organization as well as your own or/and one of the authors in our system
- please revise the legends for figures 10, 11, S6, S7, S8, and S9 so that the panels are introduced in order
- there are callouts in the manuscript text for Figure 2D, and Figure 2 doesn't have panels - please correct

A. FINAL FILES:

B. MANUSCRIPT ORGANIZATION AND FORMATTING:

Sincerely,

Reviewer #2 (Comments to the Authors (Required)):

The authors have answered all questions to my satisfaction, which markedly improved the manuscript.

Yet, there are still minor errors, which should be changed.

Please shorten the text describing the results of figure 2 without losing context. More importantly, shorten the text summarizing Suppl Fig 2, which is very long. If Suppl. Fig 2 is so important and need to be described in detail, it should not be a suppl fig.

Please improve the image quality of fig 10 (vGLUT1). Based on this representative pic, I can not see a difference, which has been stated in the graph below..

Please change the term "Having seen that Igf-I deletion disrupted the hippocampal proteome..." (page 13) to "Having seen that Igf-I deletion altered the hippocampal proteome". The word disrupted seems to overstate the phenotype.

Manuscript # LSA-2022-01691, titled “Brain IGF-I regulates LTP, spatial memory and sexual dimorphic behavior”

Point by point responses to the final Reviewer 2’s concerns.

Thank you for your positive remarks on our manuscript and please find below our responses to the minor errors you raised.

(1) Please shorten the text describing the results of figure 3 without losing context. More importantly, shorten the text summarizing Suppl Fig 2, which is very long. If Suppl. Fig 2 is so important and need to be described in detail, it should not be a suppl fig.

Following the reviewer’s comment, the text describing the results presented in Fig. 3 and Suppl. Fig. 2 has been shortened by about a third.

(2) Please improve the image quality of fig 10 (vGLUT1). Based on this representative pic, I cannot see a difference, which has been stated in the graph below.

We apologize for this. Following the Reviewer’s request we have improved the quality of the images in Fig. 10 (VGLUT1) so that the difference shown in the graph (Fig. 10C) can be better assessed.

(3) Please change the term "Having seen that Igf-I deletion disrupted the hippocampal proteome..." (page 13) to "Having seen that Igf-I deletion altered the hippocampal proteome". The word disrupted seems to overstate the phenotype.

Thank you for pointing this out, we have modified this phrase accordingly.

July 3, 2023

RE: Life Science Alliance Manuscript #LSA-2022-01691RR

Dr. Carlos Vicario
Spanish National Research Council
Molecular, Cellular biology and development
Avenida Dr. Arce 37
Madrid, Madrid 28002
Spain

Dear Dr. Vicario,

Thank you for submitting your Research Article entitled "Brain IGF-I regulates LTP, spatial memory and sexual dimorphic behavior". It is a pleasure to let you know that your manuscript is now accepted for publication in Life Science Alliance. Congratulations on this interesting work.

DISTRIBUTION OF MATERIALS:

Again, congratulations on a very nice paper. I hope you found the review process to be constructive and are pleased with how the manuscript was handled editorially. We look forward to future exciting submissions from your lab.

Sincerely,
